# Trust Your ∇: Gradient-based Intervention Targeting for Causal Discovery

**Mateusz Olko**[1,2,*]   **Michał Zając**[3,*]   **Aleksandra Nowak**[2,3,4,*]
**Nino Scherrer**[5]   **Yashas Annadani**[6]   **Stefan Bauer**[6]
**Łukasz Kuciński**[2,7]   **Piotr Miłoś**[2,8,7]
[1]Warsaw University, [2]IDEAS NCBR,
[3]Jagiellonian Univeristy, Faculty of Mathematics and Computer Science,
[4]Jagiellonian University, Doctoral School of Exact and Natural Sciences,
[5]ETH Zurich, [6]Helmholtz, TU Munich, [7]deepsense.ai,
[8]Institute of Mathematics, Polish Academy of Sciences

## Abstract

Inferring causal structure from data is a challenging task of fundamental importance in science. Often, observational data alone is not enough to uniquely identify a system's causal structure. The use of interventional data can address this issue, however, acquiring these samples typically demands a considerable investment of time and physical or financial resources. In this work, we are concerned with the acquisition of interventional data in a targeted manner to minimize the number of required experiments. We propose a novel Gradient-based Intervention Targeting method, abbreviated `GIT`, that 'trusts' the gradient estimator of a gradient-based causal discovery framework to provide signals for the intervention targeting function. We provide extensive experiments in simulated and real-world datasets and demonstrate that `GIT` performs on par with competitive baselines, surpassing them in the low-data regime.

## 1 Introduction

Estimating causal structure from data, commonly known as causal discovery, is central to the progress of science [Pearl, 2009]. Real-world systems can often be explained as a composition of smaller parts connected by causal relationships. Understanding this underlying structure is essential for making accurate predictions about the system's behavior after a perturbation or treatment has been applied [Peters et al., 2016]. Causal discovery methods have been successfully deployed in various fields, such as biology [Sachs et al., 2005, Triantafillou et al., 2017, Glymour et al., 2019], medicine [Shen et al., 2020, Castro et al., 2020, Wu et al., 2022], earth system science [Ebert-Uphoff and Deng, 2012], or neuroscience [Sanchez-Romero et al., 2019]. In machine learning, causal decomposition has been shown to enable sample-efficient learning and fast adaptation to distribution shifts by only updating a subset of parameters [Bengio et al., 2020, Scherrer et al., 2022].

*Observational data*, that is the data obtained directly from the unperturbed system, are, in general, insufficient to identify a system's causal structure and only allow to determine the structure up to the so-called Markov Equivalence Class [Spirtes et al., 2000a, Peters et al., 2017]. To overcome this limited identifiability problem, causal discovery algorithms commonly leverage *interventional* data [Hauser and Bühlmann, 2012, Brouillard et al., 2020, Ke et al., 2019], which are acquired by gathering data from an experiment perturbing a part of the system [Spirtes et al., 2000b, Pearl, 2009]. The field of *experimental design* [Lindley, 1956, Murphy, 2001, Tong and Koller, 2001] is concerned with the

---

*These authors contributed equally. Corresponding author: mateusz.olko@gmail.com

37th Conference on Neural Information Processing Systems (NeurIPS 2023).

acquisition of interventional data in a targeted manner to minimize the number of required experiments, which often requires spending a significant amount of time and physical or financial resources.

In this work, we introduce a simple and effective experimental design algorithm called Gradient-based Intervention Targeting, or GIT for short, see Figure 1. GIT can be easily combined with various gradient-based causal discovery frameworks to provide an efficient active selection of intervention targets. Our method, which is grounded in the ideas from active and curriculum learning [Settles et al., 2007, Graves et al., 2017, Ash et al., 2020], collects interventional data that induce the biggest gradient on parameters of causal structure. GIT leverages the gradient-based nature of the underlying causal discovery framework and achieves better performance than the contemporary baselines.

Our contributions include:

- We introduce GIT, which is to our knowledge, the first gradient-based intervention targeting method. Due to it's plug-and-play nature, our method can be easily combined with various gradient-based causal discovery frameworks.

- Our extensive experiments on synthetic and real-world graphs demonstrate that GIT effectively reduces the amount of interventional data needed to discover the causal structure, and performs well in the low-data regime. This makes GIT a compelling option when access to interventional data is limited.

- We provide a theoretical justification of GIT and a suite of analyses introspecting its behavior and performance.

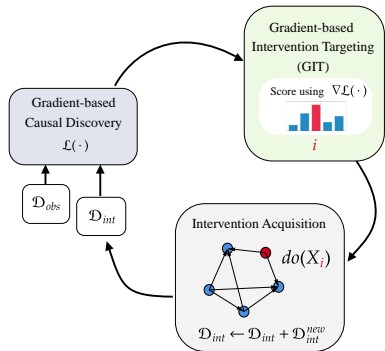

*Figure 1:* Overview of GIT's usage in a gradient-based causal discovery framework. The framework infers a posterior distribution over graphs from observational and interventional data (denoted as $\mathcal{D}_{obs}$ and $\mathcal{D}_{int}$) through gradient-based optimization. The distribution over graphs and the gradient estimator $\nabla\mathcal{L}(\cdot)$ are then used by GIT in order to score the intervention targets based on the magnitude of the estimated gradients. The intervention target with the highest score is then selected, upon which the intervention is performed. New interventional data $\mathcal{D}_{int}^{new}$ are then collected and the procedure is repeated.

## 2 Related Work

**Experimental Design / Intervention Design.** There are two major classes of methods for selecting optimal interventions for causal discovery. One class of approaches is based on graph-theoretical properties. Typically, a completed partially directed acyclic graph (CPDAG), describing an equivalence class of DAGs, is first specified. Then, either substructures, such as cliques or trees, are investigated and used to inform decisions [He and Geng, 2008, Eberhardt, 2008, Squires et al., 2020, Greenewald et al., 2019], or edges of a proposed graph are iteratively refined until reaching a prescribed budget [Ghassami et al., 2018, 2019, Kocaoglu et al., 2017, Lindgren et al., 2018]. One limitation of graph-theoretical approaches is that misspecification of the CPDAG at the beginning of the process can deteriorate the final solution. Another class of methods is based on Bayesian Optimal Experiment Design [Lindley, 1956], which aims to select interventions with the highest mutual information (MI) between the observations and model parameters. MI is approximated in different ways: AIT [Scherrer et al., 2021] uses F-score inspired metric to implicitly approximate MI; CBED [Tigas et al., 2022] incorporates BALD-like estimator [Houlsby et al., 2011]; ABCD [Agrawal et al., 2019] uses estimator based on weighted importance sampling. Although theoretically principled, computing mutual information suffers from approximation errors and model mismatches. Therefore, in this work, we explore using scores based on different principles.

**Gradient-based Causal Structure Learning.** The appealing properties of neural networks have sparked a flurry of gradient-based causal structure learning methods. The most prevalent approaches are self-supervised formulations that optimize a data-dependent scoring metric (for instance, penalized log-likelihood) to find the best causal graph $G$. Existing self-supervised methods that are capable (or can be extended) of incorporating interventional data can be categorized based on

the underlying optimization formulation into: (i) frameworks with a joint optimization objective [Brouillard et al., 2020, Lorch et al., 2021, Cundy et al., 2021, Annadani et al., 2021, Geffner et al., 2022, Deleu et al., 2022] and (ii) frameworks with alternating phases of optimization [Bengio et al., 2020, Ke et al., 2019, Lippe et al., 2022]. While structural and functional parameters are optimized under a joint objective in the former, the latter splits the optimization into two phases with separate objectives. All the aforementioned methods allow evaluation of gradient with respect to the structural and functional parameters with a batch of (real or hypothesized) interventional samples and can serve as a base framework for our proposed *gradient-based* intervention acquisition strategy.

**Gradients in Active and Curriculum Learning.** Gradients have been successfully used as a criterion to select data to process in previous work. Settles et al. [2007] introduce Expected Gradient Length (EGL), computed under the current belief, as a criterion for active learning. A batch active learning method introduced in Ash et al. [2020] also targets data points with high gradient magnitude, including uncertainty and diversity in the decision. In the area of curriculum learning, Graves et al. [2017] considers Gradient Prediction Gain (GPG), which is defined as the gradient's magnitude and is meant to be a proxy for expected learning progress. We take inspiration from those approaches to propose a novel usage of the gradient criterion in the field of causal discovery.

## 3 Preliminaries

### 3.1 Structural Causal Models and Causal Structure Discovery

Causal relationships can be formalized using structural causal models (SCM) [Peters et al., 2017]. Each of the endogenous variables $X = (X_1, \ldots, X_n)$ is expressed as a function $X_i = f_i(PA_i, U_i)$ of its direct causes $PA_i \subseteq X$ and an external independent noise $U_i$. It is assumed that the assignments are acyclic and thus associated with a directed acyclic graph $G = (V, E)$. The nodes $V = \{1, \ldots, n\}$ represent the random variables and the edges correspond to the direct causes, that is $(i, j) \in E$ if and only if $X_i \in PA_j$. The joint distribution factorizes according to

$$\mathbb{P}(X_1, \ldots, X_n) = \prod_{i=1}^{n} \mathbb{P}(X_i | PA_i). \tag{1}$$

Causal structure discovery aims to recover the ground truth graph $G$. The solution to this problem is not uniquely defined when having access only to observational data from the ground truth distribution $\mathbb{P}$. Formally, it can be determined solely up to a Markov Equivalence Class (MEC) [Spirtes et al., 2000b, Peters et al., 2017] without additional restrictive assumptions. To achieve identifiability, data from additional experiments, called interventions, need to be gathered.

A single-node intervention on $X_i$ replaces the conditional distribution $\mathbb{P}(X_i | PA_i)$ with a new distribution denoted as $\widetilde{\mathbb{P}}(X_i | PA_i)$, yielding a so-called interventional distribution:

$$\mathbb{P}_i(X) \triangleq \widetilde{\mathbb{P}}(X_i | PA_i) \prod_{j \neq i} \mathbb{P}(X_j | PA_j). \tag{2}$$

The node $i \in V$ is called the *intervention target*. An intervention that removes the dependency of a variable $X_i$ on its parents, yielding $\widetilde{\mathbb{P}}(X_i | PA_i) = \widetilde{\mathbb{P}}(X_i)$, is called hard. In this paper, we use data gathered by performing single-node interventions.

### 3.2 Online Causal Discovery and Targeting Methods

In this work, we consider an *online* causal discovery procedure outlined in Algorithm 1. Given a causal discovery Algorithm $\mathcal{A}$, the graph model $\varphi_0$ is fitted using observational data $\mathcal{D}_{obs}$. Following that, batches of interventional samples are acquired iteratively and are used by the algorithm to improve the belief about the causal structure (line 7). Intervention targets are chosen by *intervention targetting method* $\mathcal{M}$ to optimize the overall performance, taking into account the current belief about the graph structure encoded in $\varphi_{i-1}$. Below we discuss two popular choices for the method $\mathcal{M}$ (with more details deferred to Appendix D).

---

**Algorithm 1** ONLINE CAUSAL DISCOVERY

---

**input** causal discovery algorithm $\mathcal{A}$ (e.g., ENCO, see Sec 4.1), intervention targeting method $\mathcal{M}$, number of data acquisition rounds $T$, observational dataset $\mathcal{D}_{obs}$

**output** final parameters of graph model: $\varphi_T$
 1: $\mathcal{D}_{int} \leftarrow \varnothing$
 2: Fit graph model $\varphi_0$ with algorithm $\mathcal{A}$ on $\mathcal{D}_{obs}$
 3: **for** round $i = 1, 2, \ldots, T$ **do**
 4:     $I \leftarrow$ generate intervention targets using $\mathcal{M}$
 5:     $D_{int}^I \leftarrow$ query for data from interventions $I$
 6:     $\mathcal{D}_{int} \leftarrow \mathcal{D}_{int} \cup D_{int}^I$
 7:     Fit $\varphi_i$ with algorithm $\mathcal{A}$ on $\mathcal{D}_{int}$ and $\mathcal{D}_{obs}$
 8: **end for**

---

**Active Intervention Targeting (AIT)** AIT selects the intervention target according to an $F$-test inspired criterion [Scherrer et al., 2021]. It assumes that the causal discovery algorithm $\mathcal{A}$ maintains a posterior distribution over graphs (by design or using bootstrapping). To select an intervention target, a set of graphs is sampled from the posterior distribution, and interventional sample distributions are generated by intervening on each of the sampled graphs. Each potential intervention target is assigned a score by measuring the discrepancy across the corresponding interventional sample distributions.

**CBED targeting** Another approach to causal discovery is approximating the posterior distribution over the possible causal DAGs. This allows using the framework of Bayesian Optimal Experimental Design to select the most informative intervention (experiment). The score of a new experiment is given by the mutual information (MI) between the interventional data due to the experiment and the current belief about the graph structure. Hence, such an approach requires estimating MI. For instance, Causal Bayesian Experimental Design (CBED) [Tigas et al., 2022] uses a BALD-like estimator [Houlsby et al., 2011] to sample batches of interventional targets.

## 4 `GIT` method

In this work, we present a new *intervention targeting* method `GIT`. `GIT` chooses intervention targets that induce the largest update of the parameters modeling the causal structure. Inspired by *hallucinated gradients* exploited by [Ash et al., 2020] we calculate gradients on imaginary data generated by the causal model, to score possible interventions for real data acquisition.

To formally introduce our method, we first describe the requirements that need to be fulfilled by a causal algorithm $\mathcal{A}$ in order to use it with `GIT`. We then explain how `GIT` works and follow up with a discussion about causal assumptions and theoretical justification of our approach. Finally, in Section 4.1, we present a practical implementation of our method with a causal discovery algorithm $\mathcal{A}$, using a popular ENCO algorithm as an example.

**Requirements for causal discovery algorithm $\mathcal{A}$.** The intervention targeting method `GIT` can be coupled with any gradient-based causal discovery algorithm $\mathcal{A}$ (see Algorithm 1) that fulfills the following conditions:

1. $\mathcal{A}$ models a distribution over the causal DAGs, denoted by a family of probability measures $\mathbb{P}_\rho(G)$ parameterized by $\rho$, that allows sampling.
2. For each causal graph $G$, $\mathcal{A}$ maintains a corresponding family of conditional distributions, $\mathbb{P}_{G,\phi}(X_i|PA_{(i,G)})$, parametrized by $\phi$, which induces the joint distribution $\mathbb{P}_{G,\phi}$:

$$\mathbb{P}_{G,\phi}(X) \triangleq \prod_i \mathbb{P}_{G,\phi}\left(X_i|PA_{(i,G)}\right). \tag{3}$$

   If $G$ corresponds to the ground truth graph, $\mathbb{P}_{G,\phi}$ approximates the ground truth distribution over $X$.
3. $\mathcal{A}$ gives access to its loss function $\mathcal{L}$ and gradient of the loss function $\nabla_\rho \mathcal{L}$ with respect to $\rho$.

These requirements are mildly restrictive and they are fulfilled by many gradient-based discovery methods (for instance, ENCO [Lippe et al., 2022], SDI [Ke et al., 2019], DiBS [Lorch et al., 2021], DCDI [Brouillard et al., 2020] or DECI [Geffner et al., 2022]).

**Method.** `GIT` scores each possible intervention target by calculating the expected magnitude of the gradient using imaginary interventional data generated by the causal model. Gradient magnitude serves as a proxy for the size of the update that can be induced on the parameters of the causal model.

The method picks intervention that has the highest score. Formally, for a given intervention $i \in V$ we define its score $s_i$ as follows:

$$s_i \triangleq \mathbb{E}_{X \sim \mathbb{P}_{\rho,\phi,i}} \|\nabla_\rho \mathcal{L}(X)\|. \tag{4}$$

Note that the expected value is computed with the interventional distribution coming from the model, instead of ground truth, defined as:

$$\mathbb{P}_{\rho,\phi,i}(X) \triangleq \sum_G \mathbb{P}_\rho(G) \mathbb{P}_{G,\phi,i}(X). \tag{5}$$

The summation in equation 5 is taken over all DAGs and $\mathbb{P}_{G,\phi,i}$ corresponds to the joint distribution from the model for graph $G$:

$$\mathbb{P}_{G,\phi,i}(X) \triangleq \widetilde{\mathbb{P}}(X_i | PA_i) \prod_{j \neq i} \mathbb{P}_{G,\phi}(X_j | PA_{(j,G)}). \tag{6}$$

The computational procedure of `GIT`'s intervention target selection is listed in Algorithm 2. The expected value in $s_i$ is approximated using the Monte-Carlo method, see line 4 of Algorithm 2. We also use a version of Algorithm 2 where real interventional data are used in line 3 (instead of the imaginary ones from the model) and call it `GIT-privileged`. `GIT-privileged` serves as a soft upper bound in our analysis. Note however, that it is not practically useful, as collecting real interventional data would require intervention on every node in the first place.

---

**Algorithm 2** `GIT`'S INTERVENTION TARGET SELECTION

---

**input** parameters $\rho$ of distribution over graphs, functional parameters $\phi$, loss function $\mathcal{L}$, graph nodes $V$
**output** batch of intervention targets to execute: $I$
1: $\mathcal{G} \leftarrow$ sample a set of DAGs according to $\mathbb{P}_\rho(G)$
2: **for** intervention target $i \in V$ **do**
3:     $\mathcal{D}_{G,i} \leftarrow$ sample batch of data according to $\mathbb{P}_{G,\phi,i}$
4:     $s_i \leftarrow \frac{1}{|\mathcal{G}|} \sum_{G \in \mathcal{G}} \frac{1}{|\mathcal{D}_{G,i}|} \sum_{X \in \mathcal{D}_{G,i}} \|\nabla_\rho \mathcal{L}(X)\|$
5: **end for**
6: $I \leftarrow$ select a batch of targets with highest scores $s_i$

---

**Assumptions of** `GIT`. From the causal perspective, `GIT` relies exclusively on the Markov property assumption, which allows factorization of joined distribution (see Equation 1). However, `GIT` as a plug-and-play extension for causal discovery algorithms $\mathcal{A}$ inherits their assumptions. This may include, for instance, causal sufficiency or faithfulness. Our method does not require any additional assumptions on the variables $X_i$ and allows for both discrete and continuous setups.

**Theoretical justification of** `GIT`. We show the convergence of `GIT` in two contexts. First, we prove that the main setup of this paper, i.e., `GIT` with ENCO [Lippe et al., 2022], described in Section 4.1, converges. The detailed result can be found in Appendix B, but the gist of the argument is that vertices for which the model structure is not aligned with the ground truth will have non-trivial gradients and hence will be queried by `GIT`, allowing the model to improve. Moreover, we show empirically that `GIT` gradients are well correlated with the principled GPG signal of `GIT-privileged`, see Appendix F.6. Second, we show that given any convergent causal discovery algorithm, `GIT` converges if we allow a uniform sampling of intervention with small probability $\epsilon > 0$, see Appendix A. We call this approach $\epsilon$-greedy `GIT`. Importantly, on a finite sample with small enough $\epsilon$, `GIT` and $\epsilon$-greedy `GIT` are statistically indistinguishable.

### 4.1 Applicability to ENCO

We choose to use ENCO as the gradient-based causal discovery framework $\mathcal{A}$ in our main experiments (recall Algorithm 1) due to its strong empirical results and good computational performance on GPUs. ENCO maintains a parameterized distribution over graph structures, with the so-called structural parameters $\{\rho_{i,j}\}_{i,j}$ representing the adjacency matrix and a set of parameters modeling the functional dependencies, $\phi$. The structural parameters, $\rho_{i,j}$, are factorized into an edge existence parameter, $\gamma_{i,j}$, and an edge orientation parameter, $\theta_{i,j} = -\theta_{j,i}$.

The parameters are updated by iteratively alternating between two optimization phases. The goal of the first phase is to learn functions $f_{\phi_i}\left(x_i | PA_{(i,G)}\right)$, which model the conditional density of $\mathbb{P}\left(X_i | PA_{(i,G)}\right)$. The training objective is the log-likelihood loss. The second phase aims to update the parametrized edge probabilities $\rho_{i,j}$'s. To this end, ENCO collects a data sample from a mixture

*Table 1:* We count the number of setups (24), where a given method was best or comparable to the other methods (AIT, CBED, Random, and GIT; GIT-privileged was not compared against), based on 90% confidence intervals for SHD and AUSHD. Each entry shows the total count, broken down into two data regimes, $N = 1056$ and $N = 3200$, respectively, presented in parentheses.

|  | AIT | CBED | Random | GIT (ours) | GIT-privileged |
|---|---|---|---|---|---|
| mean AUSHD | 6 (2 + 4) | 6 (4 + 2) | 12 (5 + 7) | 18 (11 + 7) | 24 (12 + 12) |
| mean SHD | 10 (4 + 6) | 7 (4 + 3) | 22 (12 + 10) | 17 (10 + 7) | 24 (12 + 12) |

of interventional distributions denoted by $\mathbb{P}_I$. The graph parameters are optimized by minimizing $\mathbb{E}_{X \sim \mathbb{P}_I} L_{\text{graph}}(X)$ where:

$$L_{\text{graph}}(X) \triangleq \mathbb{E}_{G \sim P_{\gamma,\theta}} \left[ \sum_{i=1}^{n} L_G(X_i) \right], \quad L_G(x_i) \triangleq -\log f_{\phi_i} \left( x_i | PA_{(i,G)} \right), \tag{7}$$

For a detailed description of the method, distributions, and the estimators see Appendix C.1.

GIT **with ENCO details.** The loss function $\mathcal{L}$ utilized by GIT is denoted $L_{\text{graph}}$. We incorporate information from both structural parameters and use $\|\nabla_\gamma L_{\text{graph}}(X)\|^2 + \|\nabla_\theta L_{\text{graph}}(X)\|^2$ to compute the score for the intervention $i$ in line 4 of Algorithm 2. In order to sample DAGs from the current graph distribution (line 1 of Algorithm 2), we use a two-phase sampling procedure proposed in [Scherrer et al., 2021, Section 3.2] as it is scalable and guarantees DAG-ness by construction opposed to Gibbs sampling or rejection sampling techniques.

## 5 Experiments

We compare GIT against the following baselines: AIT, CBED, Random, and GIT-privileged. AIT and CBED are competitive intervention acquisition methods for gradient-based causal discovery (which we discussed in Section 3.2). The Random method selects interventions uniformly in a round-robin fashion[2]. The last approach, GIT-privileged, is the oracle method described in Section 4.

Our main result is that GIT brings substantial improvement in the low data regime, being the best among benchmarked methods for all considered synthetic graph classes and half of the considered real graphs in terms of the AUSHD metric (see Equation 9). On the remaining real graphs, our approach performs similarly to the baseline methods. Notably, in most cases, GIT surpasses MI-based approaches: CBED and AIT. We present the summary in Table 1. This result is accompanied by an in-depth analysis of the relationships between different strategies and the distributions of the selected intervention targets. Additional results in the DiBS framework [Lorch et al., 2021] with continuous data are presented in Appendix F.1.

### 5.1 Experimental Setup

We evaluate the different intervention targeting methods in online causal discovery, see Algorithm 1. We utilize an observational dataset of size 5000. We use $T = 100$ rounds, in each one acquiring an interventional batch of 32 samples. We distinguish two regimes: regular, with all 100 rounds ($N = 3200$ interventional samples), and low, with 33 rounds ($N = 1056$ interventional samples). We use $|\mathcal{G}| = 50$ graphs and $|\mathcal{D}_{G,i}| = 128$ data samples from each graph for the Monte-Carlo approximation of the GIT score. We tested different sizes of the Monte-Carlo sample and found that it does not have a major impact on performance, see Appendix F.4. For all experiments in this section we assume, following the approach of Lippe et al. [2022], that all interventions are single-node, hard, and change the conditional distribution of the intervened node to uniform.

**Datasets** We use synthetic and real-world datasets. The synthetic dataset consists of `bidiag`, `chain`, `collider`, `jungle`, `fulldag` and `random` DAGs, each with 25 nodes. The variable distributions are categorical, with 10 categories[3]. The real-world dataset consists of `alarm`, `asia`, `cancer`, `child`,

---

[2]At every step, a target node is chosen uniformly at random from the set of yet not visited nodes. After every node has been selected, the visitation counts are reset to 0.

[3]We create the datasets using the code provided by Lippe et al. [2022]. See Appendix E.1 for details.

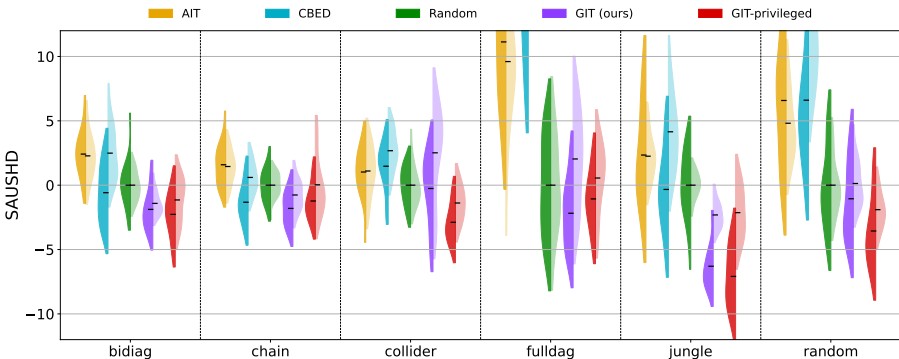

*Figure 2:* The distribution of SAUSHD (see equation 10), calculated using 25 seeds, for synthetic graphs (lower is better). The intense color (left-hand side of each violin plot) indicates the low data regime ($N = 1056$ samples). The faded color (right-hand side of each violin plot) represents a higher amount of data ($N = 3200$ samples). Note that even though the solution quality is improved when more samples are available, typically, SAUSHD is smaller in the low data regime. This is because it measures relative improvement over the random baseline, which is most visible for the small number of samples in most methods.

`earthquake`, and `sachs` graphs, taken from the BnLearn repository [Scutari, 2010]. Both synthetic and real-world graphs are commonly used as benchmarking datasets [Ke et al., 2019, Lippe et al., 2022, Scherrer et al., 2021].

**Metrics** We use the Structural Hamming Distance (SHD) [Tsamardinos et al., 2006] between the predicted and the ground truth graph as the main metric. SHD between two directed graphs is defined as the number of edges that need to be added, removed, or reversed in order to transform one graph into the other. More precisely, for two DAGs represented as adjacency matrices $c$ and $c'$,

$$\text{SHD}(c, c') := \sum_{i > j} \mathbf{1}(c_{ij} + c_{ji} \neq c'_{ij} + c'_{ji} \text{ or } c_{ij} \neq c'_{ij}). \tag{8}$$

In the experiments, we always compute SHD between the predicted and the ground truth graph. In order to aggregate SHD values over different data regimes, we introduce the area under the SHD curve (AUSHD):

$$\text{AUSHD}_m^T := \frac{1}{T} \sum_{t=1}^{T} \text{SHD}_m^t, \quad \text{SHD}_m^t := \text{SHD}(c_{gt}, c_{m,t}) \tag{9}$$

where $m$ is the used method, $T$ is the number of interventional data batches, $c_{gt}$ is the ground truth graph, and $c_{m,t}$ is the graph fitted by the method $m$ using $t$ interventional data batches. Intuitively, for small to moderate values of $T$, AUSHD captures a method's speed of convergence: the faster the SHD converges to 0, the smaller the area. For large values of $T$, AUSHD measures the asymptotic convergence. Smaller values indicate a better method. For visualizations, we use surplus of AUSHD over Random method (SAUSHD), which compares method $m$ the the Random baseline. Precisely,

$$\text{SAUSHD}_m^T := \text{AUSHD}_m^T - \mathbb{E}\left[\text{AUSHD}_{Random}^T\right], \tag{10}$$

where the expectation averages all randomness sources (e.g. stemming from the initialization). Again, smaller values indicate a better method.

### 5.2 Main Result: `GIT`'s Empirical Performance

**`GIT`'s Overall Strong Performance** We evaluate `GIT` on 24 training setups: twelve graphs (synthetic and real-world, six in each category) and two data regimes. `GIT` is the best or comparable to the baseline methods (excluding `GIT-privileged`) in 18 cases according to mean AUSHD, and 17 cases according to mean SHD, see Table 1. Additionally, `GIT` is stable, as the distribution of its AUSHD has most frequently the smallest variation among non-privileged methods (11 out of 24 cases), see Table 8 and Table 9 in Appendix F.2.2. In terms of pairwise comparison with other methods, `GIT` is better in 45 cases and comparable in 35 cases, out of a total of 96 (= 24 setups ×4

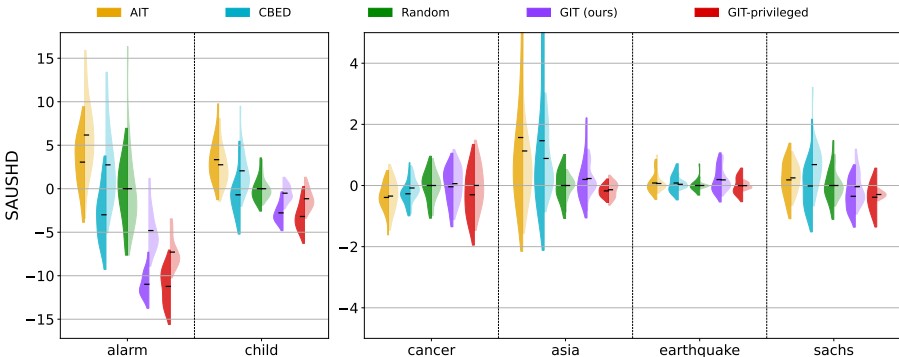

*Figure 3:* The distribution of SAUSHD (see equation 10), calculated using 25 seeds, for real-world graphs (lower is better). The intense color (left-hand side of each violin plot) indicates the low data regime ($N = 1056$ samples). The faded color (right-hand side of each violin plot) represents a higher amount of data ($N = 3200$ samples). Notice that the two plots have different scales.

other methods), see Table 7 in Appendix F.2.1. Interestingly, `GIT`'s performance for graphs with fewer nodes (`cancer`, `earthquake`) is less impressive. We hypothesize that this is because in these cases, the corresponding Markov Equivalence Class is a singleton (see Figure 4). Consequently, they require less interventional data to converge (see training curves in Appendix F.2.4), which diminishes the impact of different intervention strategies.

`GIT` **is Especially Efficient for Low Data** In the low data regime ($N = 1056$), `GIT` is better or comparable to all the other non-privileged methods for 11 out of 12 graphs, see Table 1. Pictorially, this phenomenon can be seen in Figure 2 and Figure 3, where the left-hand side of the `GIT` violin plot tends to display the most favorable behavior compared to AIT, CBED, and Random methods. This suggests that `GIT` could be a good choice when access to interventional data is limited or costly.

`GIT` **Outperforms MI-based Approaches** We also notice that the performance of MI-based approaches (CBED and AIT) is worse than the one of `GIT`, typically attaining significantly worse AUSHD (see Figure 2 and Figure 3) and SHD values (see Figure 7 and Figure 8 in the Appendix F.2). This problem is further corroborated in Section 5.3, where we show that even in the case of large interventional batch size, these methods occasionally underperform Random, unlike `GIT`, which clearly wins in such a scenario. We hypothesize the poor performance comes from approximation errors and model mismatches, subverting the MI criterion which should lead to near-optimal decisions in the case of exact mutual information computation Krause and Guestrin [2005], Nemhauser et al. [1978], Tigas et al. [2022].

`GIT` **Approximates `GIT`-privileged's Decisions** `GIT`-privileged performs the best, as it is better or comparable with all other methods for each graph and data regime (see Table 1). This strong performance is also visible in Figure 2 and Figure 3, where the mass of the method consistently occupies the favorable regions of the SAUSHD metric. These results solidify the perception of `GIT`-privileged as a soft upper-bound. Importantly, `GIT` follows it quite closely: the methods are equivalent in terms of performance in 10 cases in the low data regime, and in 5 cases in the regular data regime. Furthermore, the choices of `GIT` and `GIT`-privileged correlate highly (Spearman correlation equal 73%), see Appendix F.6. These results provide additional evidence in favor of `GIT` soundness and suggest that using data sampled from the model to compute `GIT`'s scores does not lead to severe performance deterioration. The training curves and more detailed results can be found in Appendix F.2.

## 5.3 Performance under larger interventional batch size

ENCO is sensitive to errors in the estimation of properties of interventional data. In particular, small interventional batch size may cause errors in the estimation of conditional likelihood and disrupt the causal discovery process [Lippe et al., 2022, Appendix B.2.3]. We hypothesize that those estimation errors are an important factor hindering the advantage of using our method over Random in the larger data regime. Acquiring data with small batches may result in a misaligned gradient for the model and, in consequence, in the poor assessment of the next interventional target scores.

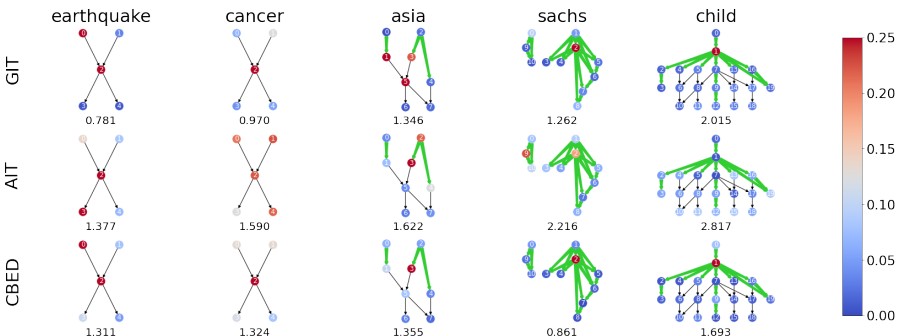

*Figure 4:* The interventional target distributions obtained by different strategies on real-world data. The probability is represented by the intensity of the node's color. The green color represents the edges for which there exists a graph in the Markov Equivalence Class that has the corresponding connection reversed. The number below each graph denotes the entropy of the distribution.

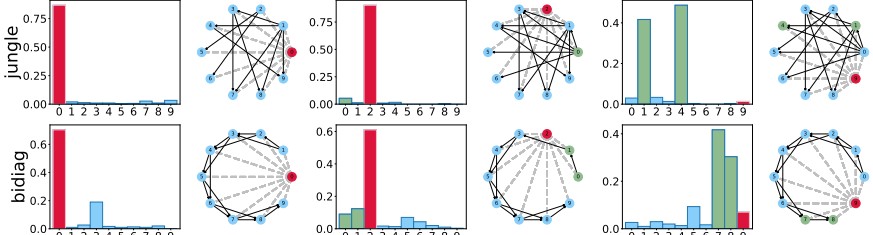

*Figure 5:* Histograms of intervention targets chosen by GIT. In this experiment, a node $v$ was chosen (denoted by a red color; $v$'s parents are indicated by green). Parameters were initialized so that the model is only unsure about the neighborhood of $v$. The solid lines denote known edges and dashed ones are to be discovered.

We perform an additional experiment in a modified regime, where each intervention yields 1024 data points instead of the previous 32. Such a regime is relevant in scenarios where setting up an intervention with a new target is costly but obtaining the individual samples is relatively cheap. We run the experiment on synthetic graphs with 25 nodes and we run for 25 acquisition rounds. We present the AUSHD values in Table 2 and

*Table 2:* Average AUSHD values (from 5 seeds) for experiments with interventional batch size equal 1024.

|          | AIT   | CBED  | Random | GIT(ours) | GIT-privileged |
|----------|-------|-------|--------|-----------|----------------|
| bidiag   | 22.6  | 17.6  | 20.4   | **16.8**  | 15.4           |
| chain    | 11.4  | 8.2   | 10.2   | **8.0**   | 7.7            |
| collider | 11.2  | 11.4  | 9.9    | **5.0**   | 4.8            |
| full     | 120.1 | 116.0 | 101.1  | **100.9** | 93.2           |
| jungle   | 22.3  | 16.7  | 19.9   | **11.4**  | 10.6           |
| random   | 38.4  | 36.2  | 32.4   | **29.9**  | 28.3           |

full SHD curves in Appendix F.3. In this setting, GIT outperforms all the standard baselines and is on par with GIT-privileged. Importantly, GIT reaches the SHD value of 0 for all graphs. Additionally, we found that GIT selects each intervention target exactly once, except for the chain graph, for which the discovery process converges already after only 15 rounds.

## 5.4 Investigating GIT's intervention target distributions

In order to gain a qualitative understanding of the GIT's behavior, we analyze the node distributions generated by respective methods on the BnLearn graphs in Figure 4. We observe that GIT often selects nodes with high out-degree, as visible in the sachs and child graphs. Intuitively, interventions on such nodes bring much information, as they affect multiple other nodes. In addition, the most frequently selected nodes in the sachs, child, and asia graphs are also adjacent to the edges for which there exists a graph in the MEC that has the corresponding connection reversed (as indicated by the green color in Figure 4). Note that in general, establishing the directionality of such an edge $(v, w)$ requires performing interventions on nodes $v, w$ (recall Section 3.1). [4]

We further explore the interventional targets and verify that GIT is able to target the most uncertain regions of the graph. In the considered setup, we select a node $v$ in the graph. Let $E_v$ be edges adjacent to $v$. We set the structural parameters corresponding to edges $e \notin E_v$ to the ground truth values and initialize in the standard way the parameters for $e \in E_v$. Such a model is only unsure

---

[4]For example, in the ENCO framework the directionality parameter $\theta_{ij}$ can only be reliably detected from the data obtained by intervening either on variable $X_j$ or $X_i$ [Lippe et al., 2022].

about the connectivity around $v$, while the rest of the solution is given. We then run the ENCO framework with `GIT` and report the intervention target distributions in Figure 5.

The interventions concentrate on $v$ (red color) and its parents (green color). This indicates the efficiency of our approach, as these are most relevant to discovering the graph structure. Indeed, to recover the solution, only the parameters for $e \in E_v$ need to be found. Intervening on $v$ changes the distributions of its descendants, providing information on the existence of edges between these variables.

## 6 Limitations and future work

- The theoretical grounding of the method involves multiple assumptions. Further work that simplifies or relaxes the assumptions and identifies fail cases would benefit the community.

- We provide proof that epsilon-greedy GIT converges with any causal discovery framework. As for pure GIT, we show its convergence only with the ENCO framework. The development of a more general theory that solidifies the approach is a promising future work direction.

- Our method can be applied in the soft-intervention case, and providing appropriate experimental evaluation would be an interesting follow-up to this work.

- Our method may need more interventions than the minimal number required to identify the causal structure. For example, GIT can be biased towards high-degree nodes, as interventions on them tend to affect a larger amount of structural parameters and result in larger gradients, which might cause suboptimal choices.

- Intervention acquisition methods (including `GIT`) seem to be less effective in a continuous setting. We believe investigating this area would benefit the community.

## 7 Conclusions

In this paper, we consider the problem of experimental design for causal discovery. We introduce a novel Gradient-based Intervention Targeting (`GIT`) method, which leverages the gradients of gradient-based causal discovery objectives to score intervention targets. We demonstrate that the method is particularly effective in the low-data regime, outperforming competitive baselines. We also provide a theoretical justification for the method and perform several analyses, confirming that `GIT` typically selects informative targets.

## Acknowledgments and Disclosure of Funding

The work of Piotr Miłoś was supported by the Polish National Science Center grant UMO-2017/26/E/ST6/00622 and UMO-2019/35/O/ST6/03464. The work of Michał Zając was supported by the Polish National Science Center Grant No. 2021/43/B/ST6/01456. The research of Michał Zając and Aleksandra Nowak has been supported by a flagship project entitled "Artificial Intelligence Computing Center Core Facility" from the DigiWorld Priority Research Area under the Strategic Programme Excellence Initiative at Jagiellonian University. We gratefully acknowledge Polish high-performance computing infrastructure PLGrid for providing computer facilities and support. Our experiments were managed using `https://neptune.ai`. We thank the Neptune team for providing us access to the team version and technical support. We thank Swedish National Supercomputing and the Berzelius Cluster for providing compute resources.

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

# Appendix

## A  Convergence of causal discovery with `GIT`

Suppose that we have some causal discovery algorithm $\mathcal{A}$ which is guaranteed to converge to the true graph in the limit of infinite data. Here we investigate if such convergence property still holds if we extend $\mathcal{A}$ with `GIT`.

Let us define $\epsilon$-greedy `GIT` as follows: every time we need to select an intervention target, we use `GIT` with probability $1 - \epsilon$, and otherwise, we choose randomly uniformly from all available targets.

**Proposition 1.** *If the causal discovery algorithm $\mathcal{A}$ is guaranteed to converge given an infinite amount of samples from each possible intervention target, then $\mathcal{A}$ with $\epsilon$-greedy `GIT` is also guaranteed to converge.*

*Proof.* Since the $\epsilon$-exploration guarantees visiting every target infinitely many times in the limit, the proof follows from the asserted convergence of $\mathcal{A}$. $\qquad\square$

*Remark* 2. ENCO with $\epsilon$-greedy `GIT` is guaranteed to converge to the true graph under the standard assumptions [Lippe et al., 2022, Appendix B.1].

*Remark* 3. Proposition 1 is asymptotic and holds for arbitrary $\epsilon > 0$. However, in a finite setup, we can choose $\epsilon$ small enough that $\epsilon$-`GIT` and `GIT` behave similarly. Our experiments show that `GIT` performs well (compared with other benchmarks) and is indistinguishable from an asymptotically convergent method.

## B  Convergence conditions of ENCO framework with `GIT`

### B.1  Preliminaries (ENCO recap)

In this section, we recall results from Lippe et al. [2022] for convergence of their causal discovery method ENCO. They formulate four theorems and a set of conditions that guarantee that the parameters of the structure converge to the true graph. For full proof and detailed explanation please refer to Appendix B in Lippe et al. [2022].

*Remark* 4. ENCO identifies common conditions for correct convergence of directionality parameters. They go as follows (see theorems B.1, B.2, and appendix B.4 in ENCO):

1. For all possible sets of parents of $X_j$ excluding $X_i$, adding $X_i$ improves the log-likelihood estimate of $X_j$ under the intervention on $X_i$, or leaves it unchanged.

$$\forall \widehat{\mathrm{pa}}(X_j) \subseteq X_{-i,j} : \mathbb{E}_{I_{X_i}, \boldsymbol{X}} \left[ \log p(X_j | \widehat{\mathrm{pa}}(X_j), X_i) - \log p(X_j | \widehat{\mathrm{pa}}(X_j)) \right] \geq 0$$

2. There exists a set of nodes $\widehat{\mathrm{pa}}(X_j)$, for which the probability to be sampled as parents of $X_j$ is greater than 0, and the following condition holds:

$$\exists \widehat{\mathrm{pa}}(X_j) \subseteq X_{-i,j} : \mathbb{E}_{I_{X_i}, \boldsymbol{X}} \left[ \log p(X_j | \widehat{\mathrm{pa}}(X_j), X_i) - \log p(X_j | \widehat{\mathrm{pa}}(X_j)) \right] > 0$$

3. For all possible sets of parents of $X_i$ excluding $X_j$, adding $X_j$ does not improves the log-likelihood estimate of $X_i$ under the intervention on $X_j$, or leaves it unchanged.

$$\forall \widehat{\mathrm{pa}}(X_i) \subseteq X_{-i,j} : \mathbb{E}_{I_{X_j}, \boldsymbol{X}} \left[ \log p(X_i | \widehat{\mathrm{pa}}(X_i), X_j) - \log p(X_i | \widehat{\mathrm{pa}}(X_i)) \right] \geq 0$$

   For at least one parent set $\widehat{\mathrm{pa}}(X_i)$, which has a probability greater than zero to be sampled, this inequality is strictly smaller than zero.

*Remark* 5. The following condition guarantees convergence of existence parameters (see theorem B.3 in ENCO):

$$\min_{\widehat{\mathrm{pa}} \subseteq \mathrm{gpa}_i(X_j)} \mathbb{E}_{\hat{I} \sim p_{I_{-j}}(I)} \mathbb{E}_{\tilde{p}_{\hat{I}}(\boldsymbol{X})} \left[ \log p(X_j | \widehat{\mathrm{pa}}, X_i) - \log p(X_j | \widehat{\mathrm{pa}}) \right] > \lambda_{sparse}$$

where $\mathrm{gpa}_i(X_j)$ is a set of nodes excluding $X_i$ which, according to the ground truth graph, could have an edge to $X_j$ without introducing a cycle, $p_{I_{-j}}(I)$ refers to the distribution over conducted interventions $p_I(I)$ excluding the intervention on variable $X_j$, and $\lambda_{sparse}$ is a positive constant.

**Theorem 6.** *(Theorem B.1 from Appendix B.4 in ENCO.) Consider the edge $X_i \rightarrow X_j$ in the true causal graph. The orientation parameter $\theta_{ij}$ converges to $\sigma(\theta_{ij}) = 1$ if the conditions from remark 4 are fulfilled.*

**Theorem 7.** *(Theorem B.2 from Appendix B.4 in ENCO.) Consider a pair of variables $X_i$, $X_j$ for which $X_i$ is an ancestor of $X_j$ without direct edge in the true causal graph. Assume all edges that appear in the true graph have converged according to theorem 6. The orientation parameter $\theta_{ij}$ converges to $\sigma(\theta_{ij}) = 1$ if the conditions from remark 4 are fulfilled.*

By Appendix B.4 from ENCO, Theorems 6, 7 hold regardless of whether we collected interventional data from node $X_i$ or $X_j$.

**Theorem 8.** *Consider an edge $X_i \rightarrow X_j$ in the true causal graph. The parameter $\gamma_{ij}$ converges to $\sigma(\gamma_{ij}) = 1$ if the condition from remark 5 holds.*

**Theorem 9.** *Assume for all edges $X_i \rightarrow X_j$ in the true causal graph, $\sigma(\theta_{ij})$ and $\sigma(\gamma_{ij})$ have converged to one. Then, the likelihood of all other edges, i.e. $\sigma(\theta_{lk}) \cdot \sigma(\gamma_{lk})$ will converge to zero under the condition that $\lambda_{sparse} > 0$.*

## B.2 `GIT`-privileged proof

We follow with proof of ENCO convergence with the GIT-privileged acquisition method under the same set of conditions from remarks 4, 5. We show that GIT-privileged collects interventional data as long as is needed for the orientation parameters to converge according to theorems 6 and 7. Then theorems 8 and 9 can be applied to show that the algorithm reached convergence.

**Assumption**    We assume that in all local minima of our loss function, the existence parameters take extreme values: $\forall_{i,j} \; \sigma(\gamma_{ij}) \in \{0, 1\}$, thus, when sufficient time for optimization is given, they stop contributing to the score. Hence in our analysis, we focus on describing only the behavior of orientation parameter gradients.

The proof is structured as follows:

1. We show that following `GIT`-privileged score allows collecting enough interventional data to direct all edges that appear in the true graph correctly, see proposition 12.

2. Then we show that, if required, additional interventional data that allows directing other edges according to theorem 7 will be collected, see proposition 13.

3. Finally, theorems 8 and 9 can be applied to show that we learned the correct graph, see proposition 14.

**Proposition 10.** *Consider the edge $X_i \rightarrow X_j$ in the true causal graph. The parameter $\gamma_{ij}$ converges to $\sigma(\gamma_{ij}) = 1$ under any set of interventions $p_I(I)$ if*

$$\min_{\hat{pa} \subseteq \mathcal{V}_{-i}} \mathbb{E}_{\hat{I} \sim p_{I_{-j}}(I)} \mathbb{E}_{\tilde{p}_{\hat{I}}(\boldsymbol{X})} \big[ \log p(X_j | \hat{pa}, X_i) - \log p(X_j | \hat{pa}) \big] > \lambda_{sparse}$$

*where $\mathcal{V}_{-i}$ is the set of all nodes excluding $X_i$, and $p_{I_{-j}}(I)$ refers to the distribution over conducted interventions $p_I(I)$ excluding the intervention on variable $X_j$.*

*Proof.* The condition guarantees that the gradient of $\gamma_{ij}$ is positive. $\qquad \square$

**Proposition 11.** *Consider the edge $X_i \rightarrow X_j$ in the true causal graph. When conditions from remark 4 are fulfilled $\|\nabla_{\theta_{ij}} L_{graph}(X_{I_i})\| = \|\nabla_{\theta_{ji}} L_{graph}(X_{I_i})\| = \|\nabla_{\theta_{ij}} L_{graph}(X_{I_j})\| = \|\nabla_{\theta_{ji}} L_{graph}(X_{I_j})\| = 0$ and the edge converged to its true value if and only if we acquired interventional data from $X_i$ or $X_j$.*

*Proof.* First, recall that ENCO does not update orientation parameters unless the interventional data was acquired from a neighboring node. Therefore, the gradient can only be zero before the intervention if the existence parameters converge to zero. This situation is guaranteed not to happen by proposition 10.

Second, when interventional data is acquired, based on the theorem 6, we know that the edge converges to its true value.

$\qquad \square$

**Proposition 12.** *Choosing intervention targets with* `GIT`*-privileged score allows collecting enough data to direct all edges that appear in the true graph properly if the following is true:*

- *For any pair of variables $X_i$, $X_j$ without a direct edge in the true causal graph, when conditions from remark 4 are fulfilled, and we acquired interventional data from either $X_i$ or $X_j$, and sufficient time for the optimization process is given, then the orientation parameters will converge to extreme values $\sigma(\theta_{ij})$, $\sigma(\theta_{ji}) \in \{0, 1\}$ or the existence parameters will converge to $\sigma(\gamma_{ij}) = \sigma(\gamma_{ji}) = 0$.*

*Proof.* The assumption and proposition 11 imply that after collecting interventional data from a node, edges that are connected to this node will not contribute to the score anymore. Thus we will not intervene on the same node twice. On the other hand, proposition 11 guarantees the score of edges that appear in the true graph will be positive until they are directed. □

**Proposition 13.** *Consider a pair of variables $X_i$, $X_j$ for which $X_i$ is an ancestor of $X_j$ without a direct edge in the true causal graph. Assume all edges that appear in the true graph has converged according to theorem 6. When conditions from remark 4 are fulfilled, and if $\|\nabla_{\theta_{ij}} L_{graph}(X_{I_i})\| = \|\nabla_{\theta_{ji}} L_{graph}(X_{I_i})\| = \|\nabla_{\theta_{ij}} L_{graph}(X_{I_j})\| = \|\nabla_{\theta_{ji}} L_{graph}(X_{I_j})\| = 0$ then the edge converged as described in theorem 7 or its existence parameters converged to 0.*

*Proof.* To zero out the gradient either orientation or existence parameters had to converge. If the orientation parameters converged we had to collect interventional data from a neighboring node (because otherwise, ENCO does not update parameters). Thus, based on the theorem 7, we know that the edge converged to its true value. □

**Proposition 14.** *Given sufficient acquisition rounds and time for optimization ENCO with* `GIT`*-privileged intervention acquisition method will recover the true graph.*

*Proof.* From propositions 12, 13 we have that `GIT`-privileged will collect interventional data from new nodes until edges that appear in the true graph are correctly directed and edges between pairs of variables $X_i$, $X_j$ without a direct edge in the true causal graph either disappear from the model or are directed according to theorem 7. By theorems 8, 9 we conclude that we indeed acquired enough interventional data to converge to the correct graph.

□

### B.3 `GIT` convergence

**Proposition 15.** *Given sufficient acquisition rounds and time for optimization ENCO with* `GIT` *intervention acquisition method will recover the true graph if the following is true:*

- *For any graph $G$ sampled from the structural belief $\mathbb{P}_\rho(\cdot)$ (recall equation 5) during* `GIT` *score estimation, the theorems and propositions from sections B.1 and B.2 hold when we use $G$ instead of the true graph and compute gradient using data from the sampled model $\mathbb{P}_{G,\phi,i}(X)$ (recall equation 6).*

*Proof.* First, note that thanks to proposition 10, if there is an edge $X_i \rightarrow X_j$ in the true graph there exists a model, that can be sampled from the belief with non-zero probability, in which this edge appears.

For each undirected edge $X_i \rightarrow X_j$ in the structural belief for which the existence parameter converged to $\sigma(\theta_{ij}) = 1$, there exists a model (with a positive probability to be sampled), that will yield a gradient of positive magnitude if and only if there is no interventional data acquired from the node connected to it. This model contains the edge $X_i \rightarrow X_j$, thus by the assumption made above and proposition 11, it will yield a positive gradient. In consequence, expectation over all possible models, when the edge is not yet directed, will yield a positive score.

Note that, when interventional data from $X_i$ or $X_j$ is acquired, the edge $X_i \rightarrow X_j$ is directed and it does not yield a gradient of positive magnitude under data sampled from $\mathbb{P}_{G,\phi,i}(X)$ for any $G \sim \mathbb{P}_\rho(\cdot)$. This stems from the fact that the gradient term zeroes out when parameter $\sigma(\theta_{ij})$ takes extreme values (0 or 1).

Hence, `GIT` score will allow to sequentially "eliminate" undirected edges. Since to update our structural belief $\mathbb{P}_\rho(\cdot)$ we use interventional data sampled from the true graph when all edges are directed, we are guaranteed (by B.2 section) that they are directed according to theorems 6, 7. Then the same argument as for `GIT`-privileged can be applied to show that we converged to the true graph. $\qquad\square$

## C  Details about Employed Causal Discovery Frameworks

### C.1  ENCO

We extend the description of the ENCO framework [Lippe et al., 2022] from Section 4.1.

**Structural Parameters.**  ENCO learns a distribution over the graph structures by associating with each edge $(i,j)$, for which $i \neq j$, a probability $p_{i,j} = \sigma(\gamma_{i,j})\sigma(\theta_{i,j})$. Intuitively, the $\gamma_{i,j}$ parameter represents the existence of the edge, while $\theta_{i,j} = -\theta_{j,i}$ is associated with the direction of the edge. The parameters $\gamma_{i,j}$ and $\theta_{i,j}$ are updated in the graph fitting stage.

**Distribution Fitting Stage.**  The goal of the distribution fitting stage is to learn the conditional probabilities $P(X_i|PA_{(i,C)})$ for each variable $X_i$ given a graph represented by an adjacency matrix $C$, sampled from $C_{i,j} \sim Bernoulli(p_{i,j})$. Note that self-loops are not allowed and thus $p_{i,i} = 0$. The conditionals are modeled by neural networks $f_{\phi_i}$ with an input dropout-mask defined by the adjacency matrix. In consequence, the negative log-probability of a variable can be expressed as $L_C(X_i) = -\log f_{\phi_i}(PA_{(i,C)})(X_i)$, where $PA_{(i,C)}$ is obtained by computing $C_{\cdot,i} \odot X$, with $\odot$ denoting the element-wise multiplication. The optimization objective for this stage is defined as minimizing the negative log-likelihood (NLL) of the observational data over the masks $C_{\cdot,i}$. Under the assumption that the distributions satisfy the Markov factorization property defined in Equation 1, the NLL can be expressed as:

$$L_D = \mathbb{E}_X \mathbb{E}_C[\sum_{i=1}^{n} L_C(X_i)]. \tag{11}$$

**Graph Fitting Stage and Implementation of Interventions.**  The graph fitting stage updates the structural parameters $\theta$ and $\gamma$ defining the graph distribution. After selecting an intervention target $I$, ENCO samples the data from the postinterventional distribution $\widetilde{P}_I$. In experiments, in the current paper, where the variables are assumed to be categorical the intervention is implemented by changing the target node's conditional to uniform over the set of node's categories. As the loss, ENCO uses the graph strcuture loss $L_{graph}$ defined in Equation 7 in the main text plus a regularization term $\lambda L_{\gamma,\theta}^{sparse}$ that influences the sparsity of the generated adjacency matrices, where $\lambda$ is the regularization strength.

**Gradients Estimators.**  In order to update the structural parameters $\gamma$ and $\theta$ ENCO uses REINFORCE-inspired gradient estimators. For each parameter $\gamma_{i,j}$ the gradient is defined as:

$$\frac{\partial L_G}{\partial \gamma_{i,j}} = \sigma'(\gamma_{i,j})\sigma(\theta_{i,j})\cdot$$
$$\cdot\mathbb{E}_{\mathbf{X},C_{-ij}}[L_{X_i \to X_j}(X_j) - L_{X_i \not\to X_j}(X_j) + \lambda], \tag{12}$$

where $\mathbb{E}_{\mathbf{X},C_{-ij}}$ denotes all of the three expectations in Equation 7 (in the main text), but excluding the edge $(i,j)$ from $C$. The term $L_{X_i \not\to X_j}(X_j)$ describes the negative log-likelihood of the variable $X_j$ under the adjacency matrix $C_{-ij}$, while $L_{X_i \to X_j}(X_j)$ is the negative log-likelihood computed by including the edge $(i,j)$ in $C_{-ij}$. For parameters $\theta_{i,j}$ the gradient is defined as:

$$\frac{\partial L_G}{\partial \theta_{i,j}} = \sigma'(\theta_{i,j})\cdot$$
$$\cdot\big(p(I_i)\sigma(\gamma_{i,j})\mathbb{E}_{I_i,\mathbf{X},C_{-ij}}[L_{X_i \to X_j}(X_j) - L_{X_i \not\to X_j}(X_j)] -$$
$$p(I_j)\sigma(\gamma_{j,i})\mathbb{E}_{I_j,\mathbf{X},C_{-ij}}[L_{X_j \to X_i}(X_i) - L_{X_j \not\to X_i}(X_i)]\big), \tag{13}$$

where $p(I_i)$ is the probability of intervening on node $i$ (usually uniform) and $\mathbb{E}_{I_i,\mathbf{X},C_{-ij}}$ is the same expectation as $\mathbb{E}_{\mathbf{X},C_{-ij}}$ but under the intervention on node $i$.

## C.2 DiBS

DiBS [Lorch et al., 2021] is a Bayesian structure learning framework which performs posterior inference over graphs with gradient based variational inference. This is achieved by parameterising the belief about the presence of an edge between any two nodes with corresponding learnable node embeddings. This turns the problem of discrete inference over graph structures to inference over node embeddings, which are continuous, thereby opening up the possibility to use gradient based inference techniques. In order to restrict the space of distributions to DAGs, NOTEARS constraint [Zheng et al., 2018] which enforces acyclicity is introduced as a prior through a Gibbs distribution.

Formally, for any two nodes $(i, j)$, the belief about the presence of the edge from $i$ to $j$ is paramerised as:

$$p(g_{ij} \mid u_i, v_j) = \frac{1}{1 + \exp(-\alpha(u_i^T v_j))} \tag{14}$$

Here, $g_{ij}$ is the random variable corresponding to the presence of an edge between $i$ to $j$, $\alpha$ is a tunable hyperparameter and $u_i, v_j \in \mathbb{R}^k$ are embeddings corresponding to node $i$ and $j$. The entire set of learnable embeddings, i.e. $\mathbf{U} = \{u_i\}_{i=1}^d$, $\mathbf{V} = \{v_i\}_{i=1}^d$ and $\mathbf{Z} = [\mathbf{U}, \mathbf{V}] \in \mathbb{R}^{2 \times d \times k}$ form the latent variables for which posterior inference needs to be performed. Such a posterior can then be used to perform Bayesian model averaging over corresponding posterior over graph structures they induce.

DiBS uses a variational inference framework and learns the posterior over the latent variables $\mathbf{Z}$ using SVGD [Liu and Wang, 2016]. SVGD uses a set of particles for each embedding $u_i$ and $v_j$, which form an empirical approximation of the posterior. These particles are then updated based on the gradient from Evidence Lower Bound (ELBO) of the corresonding variational inference problem, and a term which enforces diversity of the particles using kernels. The prior over the latent variable $\mathbf{Z}$ is given by a Gibbs distribution with temperature $\beta$ which enforces soft-acyclicty constraint:

$$p(\mathbf{Z}) \propto \exp(-\beta \mathbb{E}_{p(\mathbf{G}|\mathbf{Z})}[h(\mathbf{G})]) \prod_{ij} \mathcal{N}(z_{ij}; 0, \sigma_z^2) \tag{15}$$

Here, $h$ is the DAG constraint function given by NOTEARS [Zheng et al., 2018].

# D  Details about Intervention Targetting Methods

In this section we briefly introduce other intervention acquisition methods used for comaprison in this work.

**Active Intervention Targeting (AIT)**  Assume that the structural graph distribution maintained by the causal discovery algorithm can be described by some parameters $\rho$. Consider a set of graphs $\mathcal{G} = \{\mathcal{G}_j\}$ sampled from this distribution. AIT assigns to each possible intervention target $i \in V$ a discrepancy score that is computed by measuring the variance between the graphs $(VBG)$ and variance within the graphs $(VWG)$. The $VBG_i$ for intervention $i$ is defined as:

$$VBG_i = \sum_j \langle \mu_{j,i} - \bar{\mu}_i, \mu_{j,i} - \bar{\mu}_i \rangle, \tag{16}$$

where $\mu_{j,i}$ is the mean of all samples drawn from graph $\mathcal{G}_j$ under the intervention on target $i$, and $\mu_i$ is the mean of all samples drawn from graphs under intervention on target $i$. The variance within graphs is described by:

$$VWG_i = \sum_j \sum_k \langle [S_{j,i}]_k - \mu_{j,i}, [S_{j,i}]_k - \mu_{j,i} \rangle, \tag{17}$$

where $[S_{j,i}]_k$ is the $k$-th sample from graph $\mathcal{G}_j$ under the intervention on target $i$. The AIT score is then defined as the ratio $D_i = \frac{VBG_i}{VWG_i}$. The method selects then the intervention attaining the highest score $D_i$.

**CBED Targeting**  Bayesian Optimal Experimental Design for Causal Discovery (BOECD) selects the intervention with the highest information gain obtained about the graph belief after observing the

interventional data. Let the tuple $(j, v)$ define the intervention, where $j \in V$ describes the intervention target, and $v$ represents the change in the conditional distribution of variable $X_j$. Specifically, this means that the new conditional distribution of $X_j$ is a distribution with point mass concentrated on $v$. Moreover, let $Y_{(j,v)}$ denote the interventional distribution under the intervention $(j, v)$, and let $\psi$ denote the current belief about the graph structure (i.e. the random variable corresponding to the structural and distributional parameters $\psi = (\rho, \phi)$). BOECD selects the intervention that maximizes [Tigas et al., 2022]:

$$(j^*, v^*) = \underset{(j,v)}{\arg\max}\, I(Y_{(j,v)}; \psi \mid \mathcal{D}), \tag{18}$$

where $\mathcal{D}$ are the observational data. The above formulation necessities the use of an MI estimator. One possible choice is a BALD-inspired estimator Tigas et al. [2022], Houlsby et al. [2011]:

$$I(Y_{(j,v)}; \psi \mid \mathcal{D}) = H(Y_{(j,v)} \mid \mathcal{D}) - H(Y_{(j,v)}; \phi \mid \mathcal{D}), \tag{19}$$

with $H(\cdot; \cdot)$ denoting the cross-entropy. Note that this approach allows to select not only most informative target, but also the value of the intervention.

# E   Additional Experimental Details

## E.1   Synthetic Graphs Details

The synthetic graph structure is deterministic and is specified by the name of graph (`chain`, `collider`, `jungle`, `fulldag`), except for `random`, where the structure is sampled. Following ENCO [Lippe et al., 2022], we set the only parameter of sampling procedure, `edge_prob`, to 0.3.

The ground truth conditional distributions of the causal graphs are modeled by randomly initialized MLPs. Additionally, a randomly initialized embedding layer is applied at the input to each MLP that converts categorical values to real vectors. We used the code provided by Lippe et al. [2022]. For more detailed explanation, refer to Lippe et al. [2022, Appendix C.1.1].

## E.2   ENCO Hyperparameters

For experiments on ENCO framework we used exactly the same parameters as reported by Lippe et al. [2022, Appendix C.1.1]. We provide them in Table 3 for the completeness of our report.

*Table 3:* Hyperparameters used for the ENCO framework.

| parameter | value |
|---|---|
| Sparsity regularizer $\lambda_{sparse}$ | $4 \times 10^{-3}$ |
| Distribution model | 2 layers, hidden size 64, LeakyReLU($\alpha = 0.1$) |
| Batch size | 128 |
| Learning rate - model | $5 \times 10^{-3}$ |
| Weight decay - model | $1 \times 10^{-4}$ |
| Distribution fitting iterations F | 1000 |
| Graph fitting iterations G | 100 |
| Graph samples K | 100 |
| Epochs | 30 |
| Learning rate - $\gamma$ | $2 \times 10^{-2}$ |
| Learning rate - $\theta$ | $1 \times 10^{-1}$ |

## E.3   DiBS Hyperparameters

In Table 4, we present hyperparameters used for the DiBS framework.

## E.4   Computational Cost

We used two hardware settings, one with GPU: a single Nvidia A100, and another one with CPUs: 12 cores of Intel Xeon E5-2697 processor. In our synthetic graph experiments with ENCO on GPU, a

Table 4: Hyperparameters used for the DiBS framework.

| parameter | value |
|---|---|
| Number of particles | 20 |
| Number of particle updates | 20 000 |
| Choice of Kernel | $k([\mathbf{Z}, \Theta], [\mathbf{Z}', \Theta']) = \sigma_{\mathbf{Z}} \exp(-\frac{1}{h_{\mathbf{Z}}}||\mathbf{Z} - \mathbf{Z}'||_F^2) + \sigma_{\Theta} \exp(-\frac{1}{h_{\Theta}}||\Theta - \Theta'||_F^2)$ |
| $h_{\mathbf{Z}}$ | 5 |
| $h_{\Theta}$ | 500 |
| $\sigma_{\mathbf{Z}}$ | 1 |
| $\sigma_{\Theta}$ | 1 |
| Optimizer | RMSProp |
| Learning rate Optimizer | 0.005 |

single experiment takes on average 4 h to run, with 57 min being used by GIT to make its decisions; the rest is devoted to the underlying causal discovery algorithm (in this case, ENCO). In the case of the CPU setup, an experiment takes on average 126 h, with the GIT part taking up only 6 h. We estimate the project's overall cost to be around 50K GPUh and 2M CPUh.

# F Additional Experimental Results

## F.1 Experiments in DiBS Framework

**Experimental setup** The experimental setup closely follows the one from Tigas et al. [2022]. In the experiments, 10 batches of 10 data-points each are acquired. Each batch can contain various intervention targets. The acquisition method chooses intervention targets and values. For some of the methods, the GP-UCB strategy is used to select a value for a given intervention; see Tigas et al. [2022] for details. For every method, we run 40 random seeds. We compare the following methods:

- **Soft GIT (ours)**: gradient magnitudes corresponding to different interventions are normalized by the maximum one, then passed to the softmax function (with temperature 1). Obtained scores are used as probabilities to sample a given intervention in the current batch. GP-UCB is used for value selection.

- **Random (fixed values)**: Intervention targets are chosen uniformly randomly. The intervention value is fixed at 0.

- **Random (uniform values)**: Intervention targets are chosen uniformly randomly. The intervention value is chosen uniformly randomly from the variable support.

- **Soft AIT**: Intervention targets are chosen from the softmax probabilities of AIT scores [Scherrer et al., 2021], with the temperature 2. GP-UCB is used for value selection.

- **Soft CBED**: Intervention targets are chosen from the softmax probabilities of CBED scores [Tigas et al., 2022], with the temperature 0.2. GP-UCB is used for value selection.

The results are presented in Figure 6. We can see the performance of Soft GIT is comparable to that of Random (uniform values) in both considered graph classes. Soft AIT and Soft CBED behave similarly for Erdos-Renyi graphs, while for Scale-Free they seem to bring a small improvement.

## F.2 Performance in ENCO Framework - All Results

### F.2.1 Ranking Statistics

We present ranking statistics in Tables 5, 6, 7.

### F.2.2 AUSHD Tables

We present all AUSHD results with confidence intervals in Tables 8, 9.

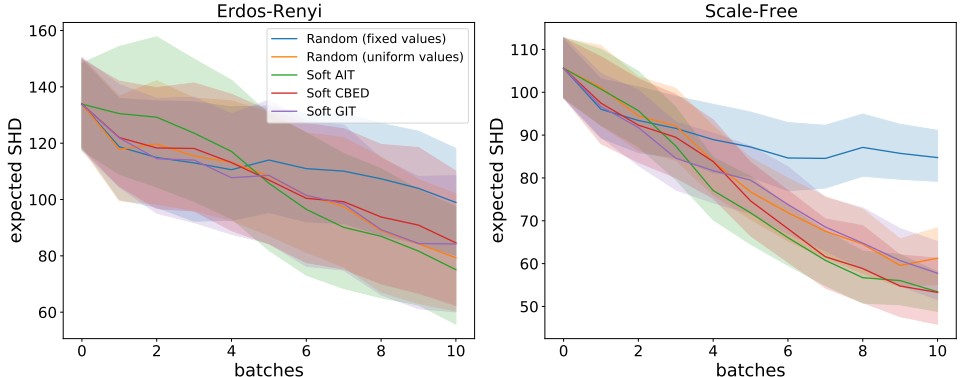

*Figure 6:* Expected SHD metric for different acquisition methods on top of the DiBS framework, for graphs with 50 nodes and two different graph classes: Erdos-Renyi and Scale-Free. 95% bootstrap confidence intervals are shown.

*Table 5:* We count the number of training setups (24), where a given method was best or at least comparable to other methods (AIT, CBED, and Random; `GIT`-privileged was not compared against), basing on 90% confidence intervals for AUSHD. Each entry shows the total count, broken down into two data regimes, $N = 1056$ and $N = 3200$ resp., presented in the parenthesis.

|  | AIT | CBED | Random | GIT (ours) | GIT-privileged |
|---|---|---|---|---|---|
| Best | $0\,(0+0)$ | $0\,(0+0)$ | $2\,(0+2)$ | $8\,(4+4)$ | $5\,(1+4)$ |
| Best or comparable | $6\,(2+4)$ | $6\,(4+2)$ | $12\,(5+7)$ | $18\,(11+7)$ | $24\,(12+12)$ |

*Table 6:* We count the number of training setups (24), where a given method was best or at least comparable to other methods (AIT, CBED, and Random; `GIT`-privileged was not compared against), basing on 90% confidence intervals for SHD. Each entry shows the total count, broken down into two data regimes, $N = 1056$ and $N = 3200$ resp., presented in the parenthesis.

|  | AIT | CBED | Random | GIT (ours) | priv. GIT |
|---|---|---|---|---|---|
| Best | $1\,(0+1)$ | $1\,(0+1)$ | $2\,(1+1)$ | $1\,(1+0)$ | $3\,(1+2)$ |
| Best or comparable | $10\,(4+6)$ | $7\,(4+3)$ | $22\,(12+10)$ | $17\,(10+7)$ | $24\,(12+12)$ |

*Table 7:* For each method we show its pairwise performance against other methods (whether it is better, comparable, or worse) based on 90% confidence intervals for AUSHD, across two data regimes ($N = 1056$ and $N = 3200$) and all twelve graphs (hence for each method there are $2 \times 12 \times 4 = 96$ pairs to consider). Each entry shows the total count, broken down into two data regimes, $N = 1056$ and $N = 3200$ resp., presented in the parenthesis.

|  | Better | Comparable | Worse |
|---|---|---|---|
| AIT | 9 (3+6) | 27 (11+16) | 60 (34+26) |
| CBED | 9 (7+2) | 35 (20+15) | 52 (21+31) |
| Random | 34 (13+21) | 36 (21+15) | 26 (14+12) |
| GIT (ours) | 45 (24+21) | 35 (21+14) | 16 (3+13) |
| GIT-privileged | 57 (25+32) | 39 (23+16) | 0 (0+0) |

### F.2.3 SHD Tables

We present SHD results for small and large data regime with confidence intervals in Tables 10, 11.

### F.2.4 ENCO - Training Curves

We provide SHD training curves for main experiments in Figures 7 and 8.

*Table 8:* AUSHD with 90% confidence intervals (in the parenthesis), for synthetic data and for low and regular data regimes ($N = 1056$ and $N = 3200$ resp.).

| | | AIT | BALD | Random | GIT (ours) | priv. GIT |
|---|---|---|---|---|---|---|
| bidiag | 1056 | 24.7 (24.1, 25.5) | 21.9 (21.1, 22.8) | 22.0 (21.5, 22.7) | 20.0 (19.5, 20.6) | 19.9 (18.6, 20.9) |
| | 3200 | 14.0 (13.0, 15.4) | 13.2 (12.5, 14.0) | 11.1 (10.5, 12.1) | 9.4 (9.0, 9.9) | 9.3 (8.0, 10.3) |
| chain | 1056 | 14.9 (14.4, 15.4) | 12.2 (11.8, 12.7) | 13.5 (13.1, 13.9) | 11.7 (11.3, 12.1) | 12.2 (11.4, 13.3) |
| | 3200 | 7.7 (7.3, 8.1) | 7.2 (6.8, 7.7) | 6.3 (6.0, 6.6) | 5.6 (5.2, 6.0) | 6.3 (5.2, 8.5) |
| collider | 1056 | 16.0 (15.2, 16.7) | 16.1 (15.5, 16.7) | 14.6 (14.1, 15.1) | 14.4 (13.4, 15.2) | 11.8 (10.9, 13.0) |
| | 3200 | 10.9 (10.2, 11.7) | 12.2 (11.6, 12.7) | 9.7 (9.2, 10.3) | 12.1 (10.9, 13.1) | 7.8 (6.9, 8.8) |
| fulldag | 1056 | 133.0 (131.2, 134.7) | 141.6 (139.1, 144.2) | 121.7 (120.4, 122.9) | 119.8 (118.7, 120.8) | 120.7 (119.1, 122.1) |
| | 3200 | 72.8 (71.0, 74.5) | 100.6 (97.8, 103.8) | 63.4 (62.0, 64.7) | 67.9 (66.0, 70.3) | 63.4 (61.2, 64.9) |
| jungle | 1056 | 23.2 (21.9, 24.6) | 20.6 (19.6, 21.7) | 20.9 (20.1, 21.7) | 14.7 (14.1, 15.4) | 13.9 (12.4, 15.5) |
| | 3200 | 11.2 (10.7, 11.9) | 13.3 (12.3, 14.3) | 9.1 (8.8, 9.5) | 6.9 (6.5, 7.2) | 6.9 (5.5, 8.3) |
| random | 1056 | 42.1 (40.5, 43.6) | 43.1 (41.5, 44.9) | 35.6 (34.6, 36.7) | 34.6 (33.7, 35.7) | 31.9 (30.4, 34.6) |
| | 3200 | 21.3 (20.4, 22.3) | 30.7 (29.0, 32.5) | 16.5 (15.8, 17.3) | 17.0 (16.3, 17.7) | 14.5 (13.6, 15.6) |

*Table 9:* AUSHD with 90% confidence intervals (in the parenthesis), for real-world data and for low and regular data regimes ($N = 1056$ and $N = 3200$ resp.).

| | | AIT | CBED | Random | GIT (ours) | priv. GIT |
|---|---|---|---|---|---|---|
| alarm | 1056 | 42.8 (41.8, 43.8) | 36.8 (35.8, 37.8) | 39.7 (38.6, 40.8) | 28.8 (28.3, 29.3) | 28.5 (27.0, 29.6) |
| | 3200 | 35.0 (33.6, 36.4) | 31.6 (30.3, 33.1) | 28.8 (27.6, 30.8) | 24.0 (23.4, 24.9) | 21.5 (20.7, 23.1) |
| asia | 1056 | 3.6 (2.9, 4.5) | 3.5 (2.8, 4.3) | 2.0 (1.8, 2.1) | 2.2 (2.0, 2.5) | 1.8 (1.7, 1.9) |
| | 3200 | 2.4 (1.9, 3.3) | 2.1 (1.9, 2.5) | 1.3 (1.2, 1.4) | 1.5 (1.4, 1.6) | 1.1 (1.0, 1.2) |
| cancer | 1056 | 2.0 (1.9, 2.1) | 2.1 (2.0, 2.3) | 2.4 (2.2, 2.6) | 2.4 (2.2, 2.5) | 2.1 (1.6, 2.6) |
| | 3200 | 1.8 (1.6, 2.0) | 2.1 (1.9, 2.2) | 2.2 (2.0, 2.3) | 2.2 (2.0, 2.4) | 2.2 (1.7, 2.6) |
| child | 1056 | 14.4 (13.7, 15.2) | 10.4 (9.6, 11.2) | 11.1 (10.7, 11.6) | 8.3 (8.0, 8.7) | 7.9 (7.0, 9.0) |
| | 3200 | 7.8 (7.1, 8.6) | 7.1 (6.5, 8.0) | 5.0 (4.7, 5.5) | 4.5 (4.2, 4.8) | 3.9 (3.2, 4.7) |
| earthquake | 1056 | 0.5 (0.4, 0.6) | 0.5 (0.4, 0.6) | 0.4 (0.3, 0.5) | 0.6 (0.5, 0.7) | 0.4 (0.2, 0.6) |
| | 3200 | 0.2 (0.1, 0.3) | 0.2 (0.1, 0.2) | 0.1 (0.1, 0.2) | 0.3 (0.2, 0.5) | 0.1 (0.1, 0.2) |
| sachs | 1056 | 3.1 (2.9, 3.3) | 2.9 (2.6, 3.1) | 2.9 (2.7, 3.1) | 2.5 (2.4, 2.7) | 2.5 (2.2, 2.8) |
| | 3200 | 1.4 (1.3, 1.6) | 1.9 (1.7, 2.2) | 1.2 (1.1, 1.3) | 1.1 (1.0, 1.3) | 0.9 (0.8, 1.0) |

*Table 10:* SHD with 90% confidence intervals (in the parenthesis), for synthetic data and for low and regular data regimes ($N = 1056$ and $N = 3200$ resp.).

| | | AIT | CBED | Random | GIT (ours) | GIT-priv. |
|---|---|---|---|---|---|---|
| bidiag | 1056 | 11.4 (10.3, 12.4) | 10.1 (9.2, 11.0) | 7.8 (7.0, 8.5) | 6.3 (5.7, 7.0) | 7.4 (6.2, 8.6) |
| | 3200 | 5.2 (4.2, 6.3) | 7.8 (6.9, 8.7) | 2.8 (2.3, 3.4) | 2.4 (1.8, 2.9) | 2.2 (0.8, 3.6) |
| chain | 1056 | 5.6 (4.8, 6.4) | 5.4 (4.6, 6.1) | 4.3 (3.8, 4.9) | 3.6 (3.0, 4.2) | 3.6 (2.0, 4.8) |
| | 3200 | 3.2 (2.6, 3.7) | 3.9 (3.4, 4.3) | 2.2 (1.7, 2.6) | 1.8 (1.3, 2.3) | 1.8 (0.2, 2.6) |
| collider | 1056 | 11.0 (10.1, 11.9) | 11.8 (11.0, 12.7) | 9.8 (9.1, 10.6) | 13.3 (12.2, 14.4) | 9.8 (7.6, 12.0) |
| | 3200 | 4.8 (3.8, 5.9) | 7.9 (6.8, 8.9) | 3.7 (2.8, 4.6) | 9.7 (7.7, 11.6) | 3.4 (1.4, 5.0) |
| fulldag | 1056 | 64.4 (61.8, 67.0) | 91.4 (86.8, 96.0) | 52.1 (50.0, 54.3) | 55.8 (53.4, 58.0) | 53.4 (49.8, 57.0) |
| | 3200 | 32.0 (30.0, 33.8) | 75.4 (71.8, 79.0) | 25.1 (22.8, 27.2) | 27.3 (25.1, 29.8) | 20.8 (19.6, 21.8) |
| jungle | 1056 | 10.4 (9.2, 11.6) | 11.6 (10.1, 13.2) | 5.7 (5.0, 6.5) | 5.1 (4.4, 5.8) | 5.2 (3.0, 7.4) |
| | 3200 | 3.5 (3.1, 3.9) | 8.3 (7.2, 9.4) | 1.9 (1.5, 2.3) | 2.2 (1.8, 2.7) | 3.0 (2.0, 4.0) |
| random | 1056 | 18.8 (17.3, 20.3) | 27.5 (25.6, 29.5) | 11.3 (10.0, 12.5) | 12.5 (11.3, 13.5) | 11.0 (9.2, 13.0) |
| | 3200 | 8.3 (7.0, 9.4) | 22.1 (19.6, 24.4) | 5.0 (4.3, 5.8) | 5.3 (4.4, 6.1) | 3.8 (2.2, 5.4) |

### F.3 ENCO - large interventional batch experiment

We provide SHD training curves for experiments with the large interventional batch in Figure 9.

### F.4 ENCO - monte carlo sampling evaluation

We provide a performance evaluation of GIT with different amount of graphs sampled from the model in Figure 10.

*Table 11:* SHD with 90% confidence intervals (in the parenthesis), for real-world data and for low and regular data regimes ($N = 1056$ and $N = 3200$ resp.).

|  |  | AIT | CBED | Random | GIT (ours) | priv. GIT |
|---|---|---|---|---|---|---|
| alarm | 1056 | 35.76 (34.04, 37.52) | 28.44 (26.68, 30.16) | 26.0 (24.71, 27.29) | 19.84 (19.0, 20.68) | 25.0 (23.2, 27.0) |
|  | 3200 | 26.15 (24.15, 28.23) | 24.33 (21.67, 27.0) | 16.0 (14.57, 17.14) | 20.0 (18.67, 21.33) | 15.2 (14.6, 15.8) |
| asia | 1056 | 2.0 (1.2, 2.68) | 1.96 (1.44, 2.4) | 0.96 (0.8, 1.12) | 1.2 (1.0, 1.36) | 1.2 (0.8, 1.4) |
|  | 3200 | 1.56 (1.12, 1.92) | 1.28 (1.0, 1.48) | 0.88 (0.79, 1.0) | 1.12 (0.96, 1.24) | 0.8 (0.6, 1.2) |
| cancer | 1056 | 1.72 (1.48, 2.0) | 2.2 (2.0, 2.4) | 2.28 (2.04, 2.48) | 2.12 (1.84, 2.4) | 2.2 (1.8, 2.4) |
|  | 3200 | 1.8 (1.6, 2.0) | 1.96 (1.72, 2.2) | 1.84 (1.6, 2.12) | 2.0 (1.76, 2.24) | 2.4 (2.0, 2.8) |
| child | 1056 | 7.32 (5.92, 8.68) | 6.36 (5.52, 7.16) | 3.52 (2.84, 4.2) | 3.72 (3.2, 4.24) | 2.8 (1.4, 4.0) |
|  | 3200 | 3.2 (2.56, 3.8) | 4.68 (3.8, 5.48) | 1.04 (0.7, 1.35) | 2.16 (1.8, 2.52) | 1.8 (0.4, 3.0) |
| earthquake | 1056 | 0.12 (0.0, 0.2) | 0.12 (0.0, 0.2) | 0.0 (0.0, 0.0) | 0.24 (0.08, 0.36) | 0.0 (0.0, 0.0) |
|  | 3200 | 0.04 (-0.04, 0.08) | 0.0 (0.0, 0.0) | 0.0 (0.0, 0.0) | 0.2 (0.08, 0.32) | 0.0 (0.0, 0.0) |
| sachs | 1056 | 0.84 (0.68, 1.0) | 1.28 (0.96, 1.6) | 0.6 (0.4, 0.8) | 0.52 (0.32, 0.72) | 0.4 (0.0, 0.8) |
|  | 3200 | 0.48 (0.32, 0.64) | 1.48 (1.16, 1.76) | 0.24 (0.08, 0.36) | 0.48 (0.28, 0.68) | 0.0 (0.0, 0.0) |

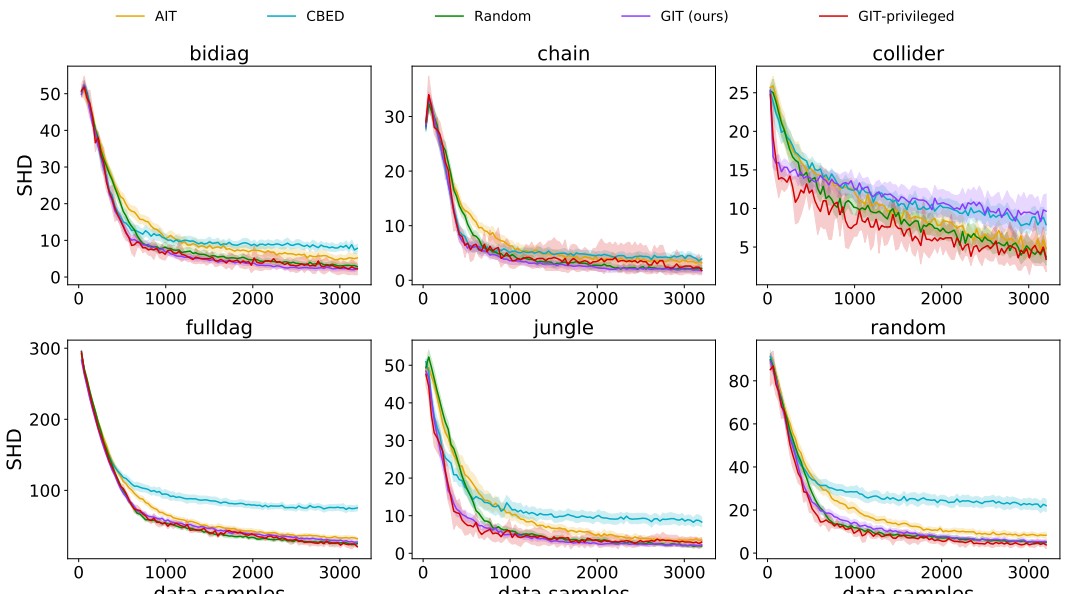

*Figure 7:* Expected SHD metric for different acquisition methods on top of the ENCO framework, for synthetic graphs with 25 nodes. 95% bootstrap confidence intervals are shown.

## F.5 ENCO - Large Synthetic Graphs

In order to study the scalability of our method, we perform an additional evulation on selected synthetic graphs in which we increase the number of nodes to 100. We comapre the performance of different acquistion methods used with ENCO in Figure 11. We observe that `GIT` exhibits very good results, converging to lower SHD values with significantly less acquistion steps compared to all the other methods. This confirms the superiority of `GIT`, even within a larger graph regime.

## F.6 ENCO - Correlation Scores

In Figure 12, we present the correlation of scores of the tested targeting methods. Importantly, the high correlation of `GIT` and `GIT-privileged` supports the hypothesis that imaginary gradients are a credible proxy of the true gradients and thus validates `GIT`. Otherwise, correlations are relatively small, suggesting that the studied methods use different decision mechanisms. Understanding this phenomenon is an interesting future research direction.

Below, we provide more details about computing the correlations. Let us denote by $s_{b,i}^m$ the score produced by method $m$ for the batch $b$ and the node $i$. In order to eliminate effects such as changing

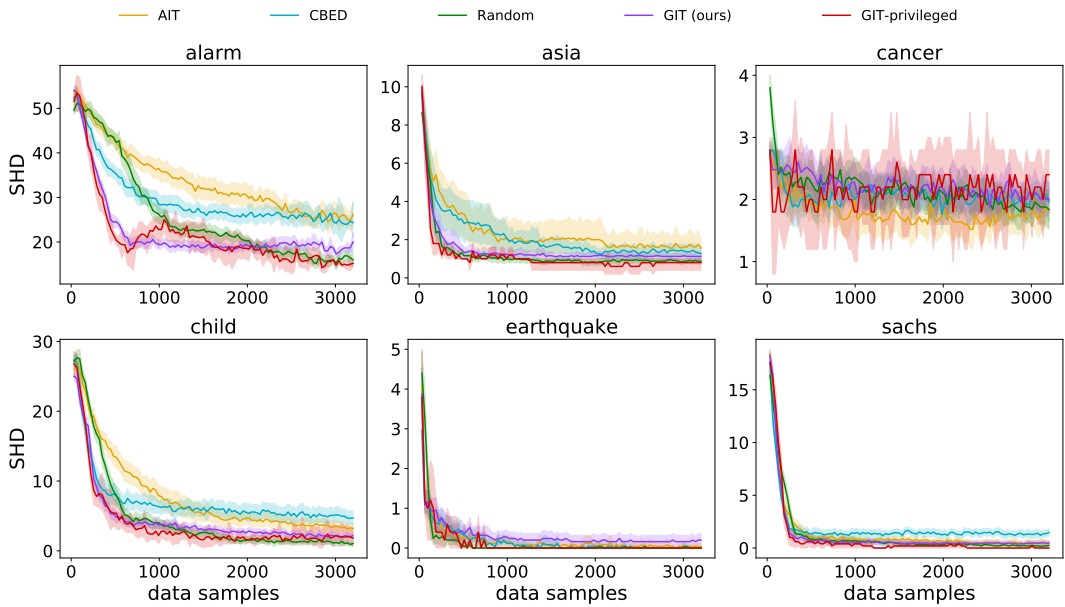

*Figure 8:* Expected SHD metric for different acquisition methods on top of the ENCO framework, for graphs from BnLearn dataset. 95% bootstrap confidence intervals are shown.

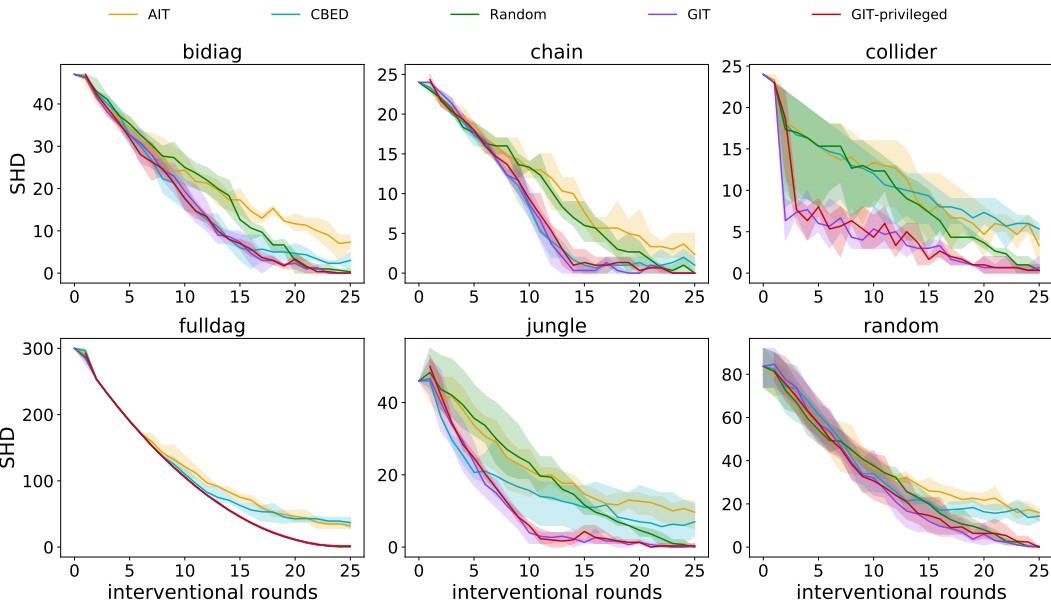

*Figure 9:* Expected SHD metric for GIT with an interventional batch size of 1024 samples. 95% bootstrap confidence intervals are shown, and results were computed using 3 random seeds.

scores scales during the discovery process, we normalize the scores as $\bar{s}_{b,i}^m := \frac{s_{b,i}^m}{\sum_{j=1}^N s_{b,j}^m}$. For every pair of methods $m, m'$ and node $i$, we compute Spearman's rank correlation score $r_s\left(\bar{s}_{\cdot,i}^m, \bar{s}_{\cdot,i}^{m'}\right)$. We average over the nodes to get the scalar correlation value $\text{corr}(m, m') := \frac{\sum_{j=1}^N r_s(\bar{s}_{\cdot,i}^m, \bar{s}_{\cdot,i}^{m'})}{N}$.

In addition, we present Pearson's correlations in Figure 13. Conclusions from the analysis of the Spearman's rank correlation hold; in particular, the correlation between `GIT` and `GIT`-privileged is high.

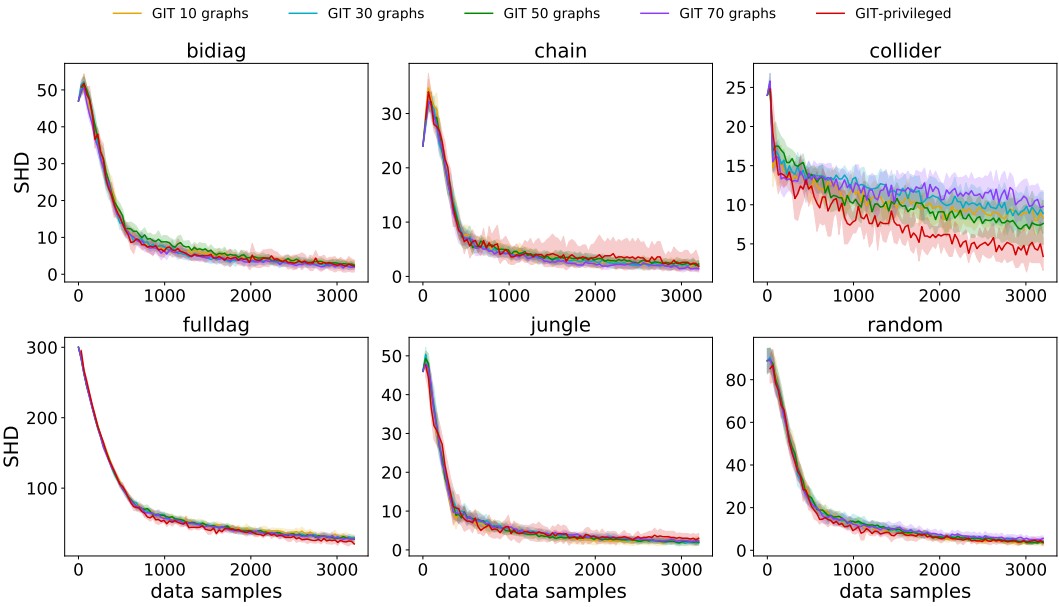

*Figure 10:* Expected SHD metric for GIT with different numbers of graphs samples used to estimate score for interventions (see line 2 in Algorithm 2). 95% bootstrap confidence intervals are shown, results were computed using 10 random seeds.

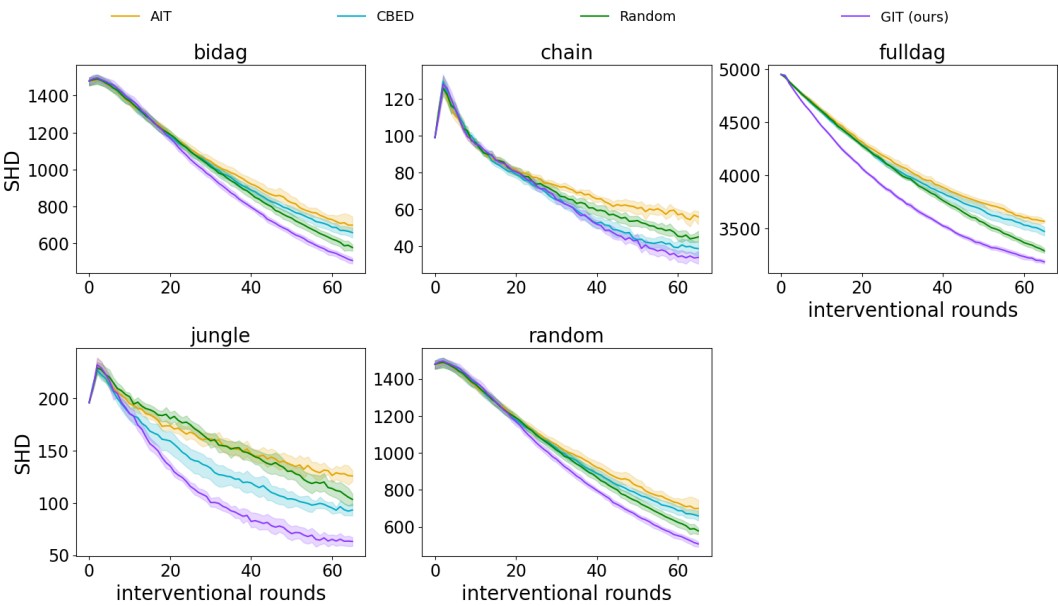

*Figure 11:* Expected SHD metric for different acquisition methods on top of the ENCO framework, for synthetic graphs with 100 nodes. 95% bootstrap confidence intervals are shown.

## F.7 ENCO - Intervention Targets Distribution

In this section, we provide additional histograms and plots with regard to the interventional target distributions obtained by different intervention methods as discussed in Section 5.4 in the main text.

In Figure 14, we present the histograms of the target distributions for the real graphs for each of the intervention acquisition methods. Note that those histograms represent the same information as the node coloring in Figure 4. It may be observed that the distributions obtained by GIT concentrate on fewer nodes than those obtained by the AIT and CBED approaches. The only exceptions being the

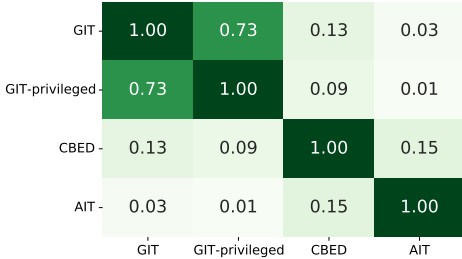

*Figure 12:* Spearman's rank correlation of the scores produced by different acquisition methods, averaged over nodes.

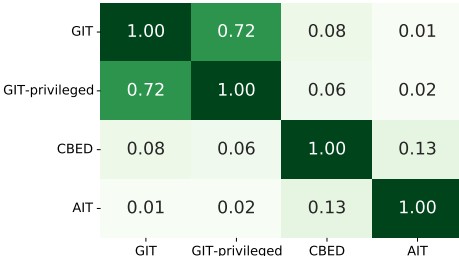

*Figure 13:* Pearson correlation of the scores produced by different acquisition methods, averaged over nodes. We can see similar trends as in the case of Spearman's rank correlation, in particular, a high correlation of `GIT` and `GIT`-privileged.

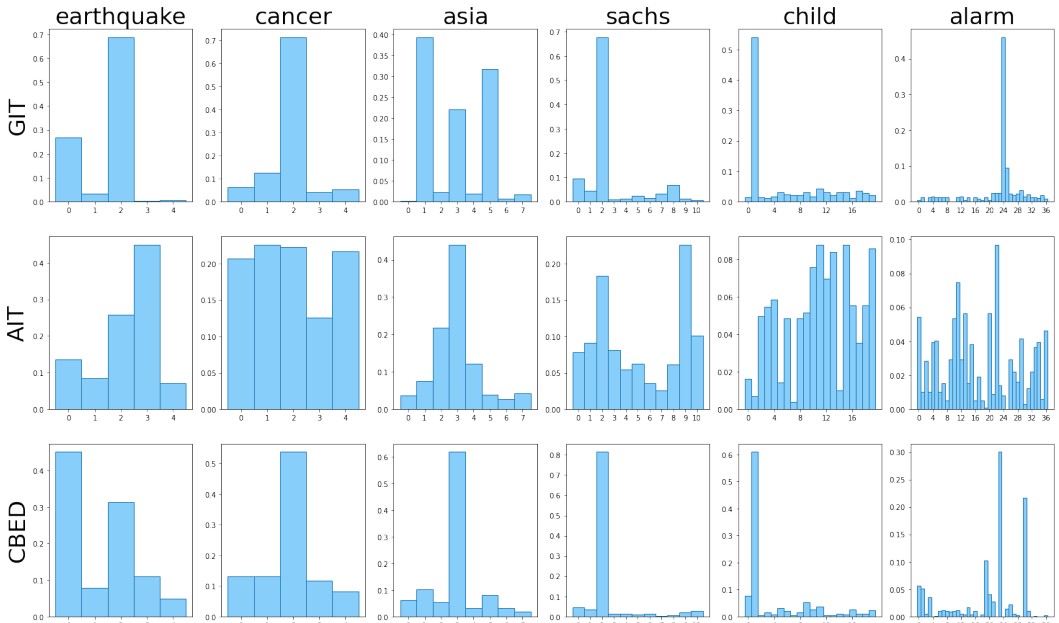

*Figure 14:* The histograms of chosen interventional targets in all data acquisition steps for different strategies computed on the real-world data.

`sachs` and `child` datasets, for which the entropy of CBED approach is smaller (recall Figure 4). Note, however, that CBED underperforms on those graphs (recall Figure 3 in the main text or see Figure 8). This is in contrast to `GIT`, which maintains good performance.

Finally, in Figure 15, we present the interventional target distribution on the `alarm` graph. We observe that each method intervenes on at least one node incident to the critical edges in the Markov

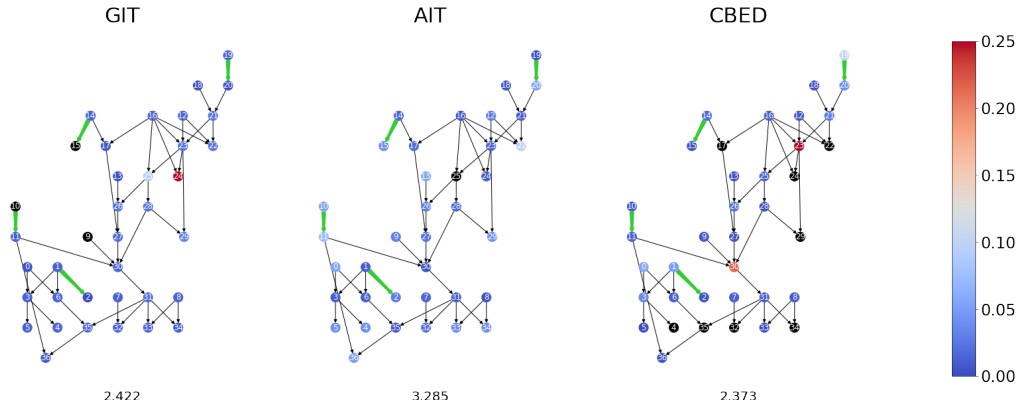

*Figure 15:* The interventional target distribution for the alarm graph. The green color represents the edges for which there exists a graph in the Markov Equivalence Class that has the corresponding connection reversed. Black color is used to indicate node for each no data is collected. We may observe that each method intervenes on at least one node incident to the critical edges. However, both AIT and CBED do not converge for this dataset and struggle to achieve good results.

Equivalence Class (as indicated by the green color in the plot). However, both AIT and CBED struggle to achieve convergence and suffer low performance, as can be observed in Figure 8.

### F.7.1    ENCO - Obtained Synthetic Graphs

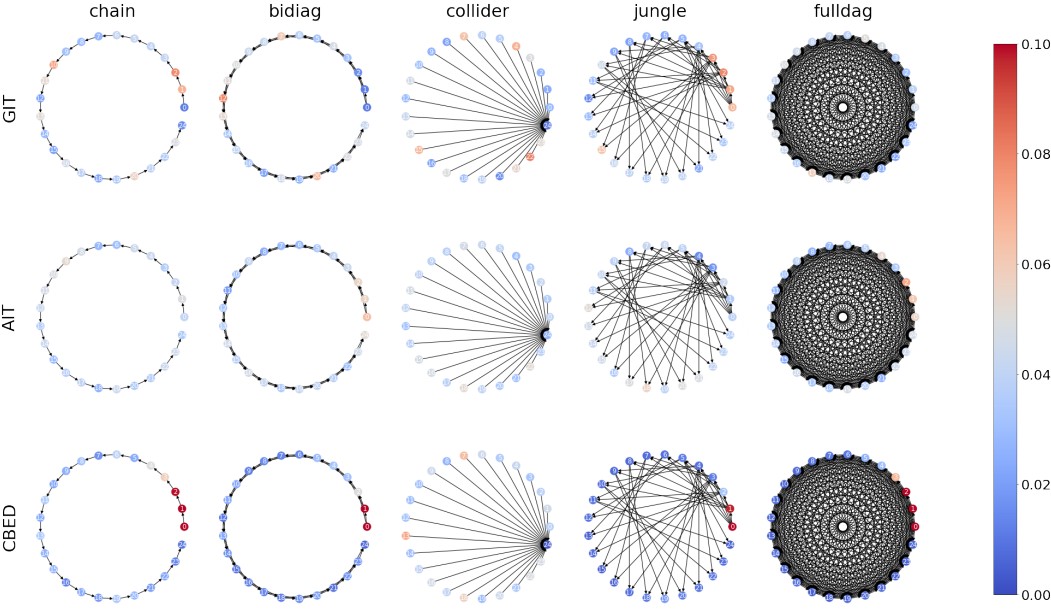

*Figure 16:* The interventional target distributions obtained by different strategies on synthetic data. The probability is represented by the intensity of the node's color. For clarity of the presentation, we choose not to color the critical edges in the corresponding Markov Equivalence Classes. This is because *all* edges of all the presented graphs would need to be colored. The only exception is the collider graph, which is alone in its Markov Equivalence Class.

In addition, we present the results obtained for the synthetic graphs in Figure 16 and the corresponding histograms in Figure 17. Note that in this case the results are also averaged by different ground truth distributions, which means that any regularities in selecting the nodes come rather from the graph structure than from data distribution.

Interestingly, we may observe that for the `jungle` and `chain` graphs GIT often intervenes on the nodes which are the first ones in the topological order (as indicated by low node numbers in the plots). This is again intuitive, as intervening on those nodes can impact more variables lower in the hierarchy. In addition, note that for the chain graph, knowing its Markov Equivalence Class and setting the directionality of an edge automatically makes it possible to determine the directionality of edges for all subsequent nodes in the topological order. Hence intervening on the nodes which are the first ones in the ordering may convey more information and is desired.

We may also observe that the CBED seems to focus only on the first nodes in the topological order, despite the data distribution, which in some graphs (as the `chain` graph) may be desired, but in others seems to be an oversimplified solution. Note that CBED often struggles to converge – this may be observed in Figure 7.

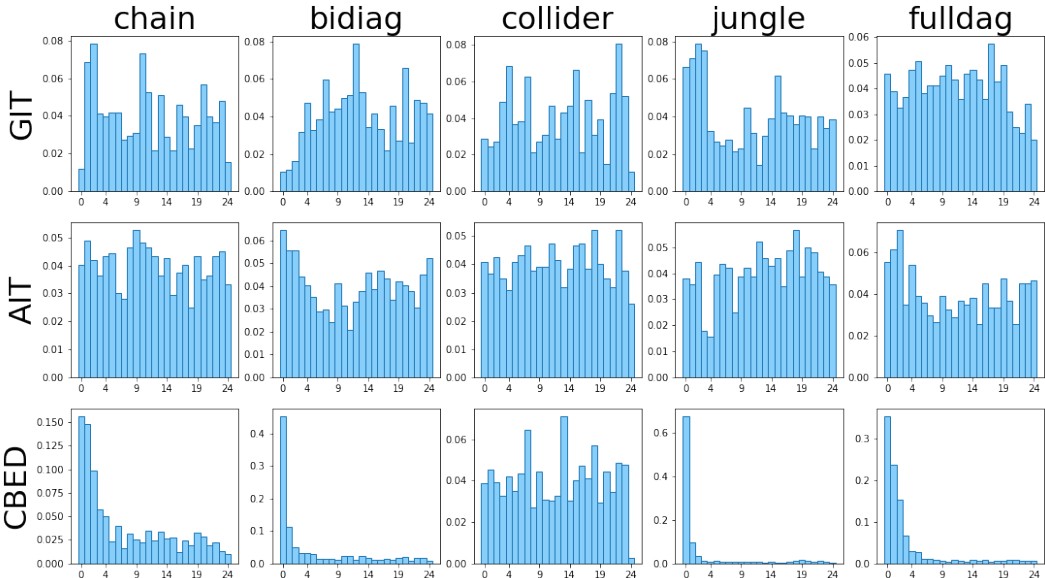

*Figure 17:* The histograms of chosen interventional targets in all data acquisition steps for different strategies computed on the synthetic data.

### F.7.2   Discussion on Small Real-World Graphs

We provide a more detailed discussion on the differences between the `earthquake` and `cancer` graph distributions and the way it affects the GIT method.

Consider Figure 4 in the main text. Note that the middle node in the `earthquake` graph corresponds to setting off a burglary alarm, an event very unlikely to happen in observational data but which, when occurs, triggers a change in the distributions of the nodes lower in the hierarchy (see the conditional distributions in Table 12). The chance of starting an alarm is also very high in case a burglary has happened (the left-most node in the graph). Hence the GIT concentrates on those two nodes as they have the largest impact on the entailed distribution.

A similar situation can be observed for the `cancer` graph, where the middle node corresponds to a binary variable indicating the probability of developing the illness. Even though the two parents of the cancer variable (pollution and smoke, represented by nodes 0 and 1, respectively) share a causal relationship with cancer, their impact on the cancer variable is limited. In other words, the chances of developing cancer, no matter whether being subject to high or low pollution or being a smoker or not, remain rather small (see the conditional distributions for cancer variable in Table 13). Hence, the only way in which one can gather more information about the impact of having cancer on the distributions of its child variables (nodes 3 and 4) is by performing an intervention. In consequence, it may be observed that GIT prefers to select nodes that allow to gather data that otherwise would be hard to acquire in the purely observational setting.

*Table 12:* The conditional distribution in the `earthquake` graph.

| Variable | Parents | Values | Distribution |
|---|---|---|---|
| Burglary | – | [True, False] | $[0.01, 0.99]$ |
| Earthquake | – | [True, False] | $[0.02, 0.99]$ |
| Alarm | Burglar=True, Earthquake=True | [True, False] | $[0.95, 0.05]$ |
| Alarm | Burglar=False, Earthquake=True | [True, False] | $[0.29, 0.71]$ |
| Alarm | Burglar=True, Earthquake=False | [True, False] | $[0.94, 0.06]$ |
| Alarm | Burglar=False, Earthquake=False | [True, False] | $[0.001, 0.999]$ |
| John Calls | Alarm=True | [True, False] | $[0.9, 0.1]$ |
| John Calls | Alarm=False | [True, False] | $[0.05, 0.95]$ |
| Mary Calls | Alarm=True | [True, False] | $[0.7, 0.3]$ |
| Mary Calls | Alarm=False | [True, False] | $[0.01, 0.99]$ |

*Table 13:* The conditional distribution in the `cancer` graph.

| Variable | Parents | Values | Distribution |
|---|---|---|---|
| Pollution | – | [Low, High] | $[0.9, 0.1]$ |
| Smoker | – | [True, False] | $[0.3, 0.7]$ |
| Cancer | Pollution=Low, Smoker=True | [True, False] | $[0.03, 0.97]$ |
| Cancer | Pollution=High, Smoker=True | [True, False] | $[0.05, 0.95]$ |
| Cancer | Pollution=Low, Smoker=False | [True, False] | $[0.001, 0.999]$ |
| Cancer | Pollution=High, Smoker=False | [True, False] | $[0.02, 0.98]$ |
| Xray | Cancer=True | [True, False] | $[0.9, 0.1]$ |
| Xray | Cancer=False | [True, False] | $[0.2, 0.8]$ |
| Dyspnoea | Cancer=True | [True, False] | $[0.65, 0.35]$ |
| Dyspnoea | Cancer=False | [True, False] | $[0.3, 0.7]$ |

## F.8 ENCO - Experiments with Pre-Initialization

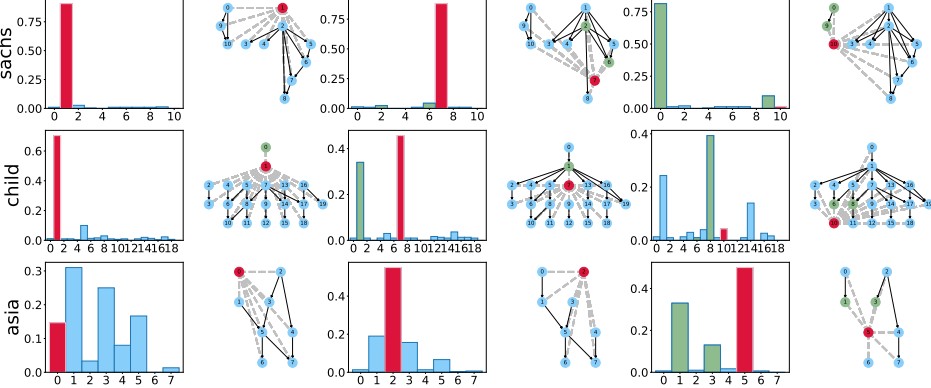

*Figure 18:* Histograms of intervention targets chosen by `GIT`. The red color corresponds to the selected node, while the green color indicates the node's parents. The edges on which standard initialization was used are indicated by gray dashed lines. The rest of the solution is given in the initialization.

In addition to the discussion on the target distributions in the case of pre-initializing parts of the graph with the ground truth solution (presented in the main text for synthetic graphs in Section 5.4), we present results of the same experiment computed on the real-world graphs. The results are presented in Figure 18.

Similarly as for the synthetic graphs, here we also observe that the GIT concentrates either on the selected node $v$ or on its parents (denoted respectively by red and green colors in the plots).

