# OpenReview forum: "Trust Your $\nabla$: Gradient-based Intervention Targeting for Causal Discovery"
_NeurIPS.cc/2023/Conference — NeurIPS 2023 poster_

### Official Review · Reviewer_zVWz · 2023-06-26

**Soundness:** 3 good
**Presentation:** 3 good
**Contribution:** 3 good
**Rating:** 5
**Confidence:** 3

**Summary:**

The paper proposes to exploit the gradient information to efficiently select intervention targets. The novel approach is called Gradient-based Information Targeting (GIT), it is claimed that it is the first gradient-based intervention targeting method. A number of realistic benchmarks are used to validate the approach.

**Strengths:**

The approach is sound, and it is reported that it performs quite well.

**Weaknesses:**

The idea to make use of the gradient information is not new. The actual method relies on the scores (eq. 4) that are proportional to the expected magnitude of the gradient. Although it might be novel to apply the gradient information to identify the nodes for interventions, the gradient magnitude information was already well studied both in a number of theoretical and practical publications.


**Questions:**

I would like a clarification: the parameters of the model are associated with graph edges. So, the gradient information is the information on the edges. Once a batch of targets with the highest scores are identified, what information (observation) is added exactly? If I remember, the BNlearn generates data as a matrix: number of observations (rows) \times variables (nodes, in columns). So, if you decide that you need more information on a particular node (column), how can you add such a partial information? Aren't you going to get missing values for other nodes in this case?

---

> ### Author Rebuttal · Authors · 2023-08-09
>
> We thank the Reviewer for the feedback. We are glad that the Reviewer appreciates that GIT is sound and experiments are well tailored for the problem. Below we provide answers to the Reviewer's questions and concerns:
>
> 1. [Lack of novelty]- It is true that the idea of using gradient signal in guided data selection has been proven to be fruitful in many contexts throughout machine learning, especially in active and curriculum learning. However, it is not immediately clear how to apply these ideas to intervention targeting, and even if so, there still is a gap between expectations and how to achieve them. Our work bridges this gap: we are the first to associate gradients of the structural loss used by causal discovery methods with intervention targeting. We show how to bring all the necessary components together in a simple manner that achieves strong empirical performance and is agnostic to the underlying gradient-based causal discovery framework. GIT is accompanied by a theoretical analysis, which underpins its validity. We believe all this is a proper scientific contribution with value to the community.
>
> *Questions:*
> 1. [Clarification of interventional data acquisition] - Once a target node with the highest score is identified, an intervention is performed on that node (in the real, underlying model). This effectively changes the joint distribution of the data (see Equation 2 in our paper and lines 107-116). We sample observations of all random variables in the graph from this *new* distribution. We then add this new information to the set of interventional data $D_{int}$ (see Algorithm 1) and retract the intervention in the real model. BnLearn apart from datasets provides also models which allow conducting such operations and collecting arbitrary large datasets.
>
> Please let us know if the above answers the Reviewer’s comments and questions. We would also be more than happy to address any other suggestions. In case there are none, and given the Reviewer’s positive outlook on soundness, presentation, and contribution, we would gently ask the Reviewer to consider increasing the score.

---

> > ### Comment · Reviewer_zVWz · 2023-08-12
> >
> > I acknowledge that I read the authors response.

---

> > > ### Author Response · Authors · 2023-08-12
> > >
> > > We thank the Reviewer for reading our response and their previous constructive feedback that further helped to improve our work. We would be happy to answer any open questions and, otherwise, would kindly ask the Reviewer to consider raising their score.

---

### Official Review · Reviewer_E5qP · 2023-07-01

**Soundness:** 3 good
**Presentation:** 3 good
**Contribution:** 3 good
**Rating:** 6
**Confidence:** 2

**Summary:**

This paper proposes a method for efficient acquisition of interventional data. Specifically, given observational data and causal discovery algorithm A, the authors propose a method (GIT: Gradient-based Intervention Targeting) that chooses particular intervention(s) to enrich the data and enhance the causal discovery process. The norm of  the gradient (of the structural loss) is used as the acquisition function.

The paper indeed tackles an important problem (causal discovery with active experiment design), with no apparent restrictions on the random variables (though to varying degrees of success in the empirical evaluation). The paper provides a fair number of empirical evaluations and compares the method to a decent number of baselines.


**Strengths:**

1. The paper tackles an important problem, and provides a clean solution that seems to be  compatible with many causal discovery frameworks.
2. The paper is generally well-written and easy to digest.
3. Extensive empirical evaluations are provided which shows that the proposed method is superior especially in the low-data regime.

**Weaknesses:**

1. I believe the main weakness of the paper is that I do not fully get why the gradients estimated from “ imaginary” interventional data approximate those you get from “real” interventional data. It would be great to:
(i) give a hypothesis for why this is the case;
and/or (ii) state clearly (in the main text) that it is indeed surprising and is an interesting direction of future work.
I think this is an interesting phenomenon that should be highlighted more in the text.

2. There is a clear improvement of GIT over baselines; however, I find it surprising that the random baseline is doing much better than other baselines. You state that this is due to approximation errors and model mismatches. Can you elaborate? Is it safe to conclude that in this setting, Random is the SOTA method (other than GIT)?
3. While the framework seems generic and not specific to categorical variables, the experiments suggests that GIT have minimal advantage over Random in the continuous settings. Figure 6 suggests that GIT only outperform Random-fixed (which I’m not sure is a reasonable baseline).
3. Nitpicks:

- Line 291: section 5.3
- Line 300: Did you mean SAUSHD instead of EAUSHD?
- Result in A is not really specific to GIT; I believe it is true for all algorithms.



**Questions:**

1. I like the results where you show the correlation of the scores produced by different acquisition methods. However, I think further exploration here would be valuable. For example:
- How the correlation increases/decreases with increasing/decreasing number of samples in the Monte Carlo approximation ($|\mathcal{D}_{G,i}|$). This may help understand the gap between the privileged and simulated gradients (i.e., does approximation error play a big role?).
- Is the correlation in earlier batches lower than that of later ones? And by how much? I suspect the answer to the first question is yes since for later batches the estimated DAG is "closer" to the true DAG.

2. I'd be interested to see an evaluation of the performance of \epsilon-greedy GIT. My presumption is that it could serve as a useful safeguard against possible inaccuracies in gradient estimation or discrepancies with the "real" privileged gradient.
3. I'd like to understand the rationale behind choosing Random-fixed as a baseline in the experiments under the DiBS Framework. From my viewpoint, this baseline selection seems somewhat arbitrary. Could you elaborate on why you chose this specific baseline?
4. In algorithm 2, you choose a set of interventions in each round (32 as stated in section 5). Do these correspond to the number of data samples generated? Or are they the number of nodes intervened on? If the latter, does that mean only one point is generated per intervention? would increasing the number of samples per intervention help?
5. From a quick read of appendix B, my understanding leads me to believe that these theorems are specific to GIT-Privileged and not GIT with “imaginary” interventional samples. Could you confirm if my interpretation is correct or if I may have misunderstood the points made?


**Limitations:**

No negative societal impact is expected.

---

> ### Author Rebuttal · Authors · 2023-08-09
>
> We appreciate the Reviewer's feedback. It's gratifying to know that the Reviewer recognizes GIT as a robust solution to the crucial problem of intervention targeting, and acknowledges its compatibility with various causal discovery frameworks. We're also pleased that the Reviewer acknowledges the thoroughness of our empirical evaluation. Below, we address the raised concerns:
>
> 1. [“Imaginary” gradients approximate the “real” ones] - Consider an algorithm (e.g., ENCO) that throughout the training consecutively learns to better approximate the real solution. In consequence, the “imaginary” data sampled from such a model progressively become a better approximation of the real interventional data. In Appendix F.5 we also empirically observe that the correlation between the scores of GIT and GIT-privilege is high. Note that the approach of using “imaginary” gradients is also well-grounded in the active learning literature (Ash, Jordan T., et al. 2019). At the same time, we agree that the properties of this process are far from being understood. We add this point to the future work section.
>
> 2. [Strength of Random Baseline] - indeed, at first this came as a surprise for us as well, and, in our view, reinforces the need for theoretical understanding of the fail cases of those methods. We suspect that the observed behavior stems from the poor quality of approximations used internally by AIT and CBED. For example, MI used in CBED might be susceptible to errors in highly dimensional situations.
> Note, however, that random selection is a strong baseline in some active learning scenarios. For instance, (Lowell et al. 2018) observe that an actively-acquired dataset does not consistently outperform training on i.i.d sampled data.
> 3. [Continuous variable setup and Figure 6] - Thank you for bringing up this point. Indeed, we only show that GIT is applicable to the continuous situation but not necessarily effective. We now add it to the limitations. Note that in this continuous setup, all the baseline methods (not only GIT) perform very similarly (except for the Random-fixed). We speculate that in the continuous case, it is perhaps more important to identify the values of the nodes than properly select the intervention nodes (consider the difference between Random-Fixed and Random-uniform in Figure 6 in our Appendix, and Figure 3a/b in Tigas et al. 2022). A detailed explanation is an interesting direction for future work.
>
> Questions:
>
> 1. [Further exploration of correlations] - Following the Reviewer's suggestion, we analyzed how correlations change over time. The results are shown in Figure R.3 in the Rebuttal PDF. Surprisingly, we noticed that although the correlation between GIT and GIT-privileged remains high, the overall correlation between different methods decreases as the number of acquired batches increases. We hypothesize that early in the training process, identifying the best target nodes is relatively easy, leading to similar solutions across methods. However, as the number of steps increases, finding nodes that significantly improve performance becomes more challenging (this idea is also supported by the initial rapid SHD convergence shown in Figure 7 in the Appendix).
>
> 2. [Evaluation of \epsilon-greedy GIT] - Thank you for your suggestion. We evaluated GIT, random, and $\epsilon$-GIT ($\epsilon$=0.33) on collider and jungle graphs. The results are presented in Figure R.2 in the rebuttal PDF. Interestingly, for the jungle graph (where GIT performed well, as shown in Figure 2 of the main paper), GIT and $\epsilon$-GIT exhibit similar performance. Additionally, as pointed out by the Reviewer, $\epsilon$-GIT seems to enhance performance in cases where GIT struggles due to inaccuracies or disparities with the true gradient. This is evident from the performance on the Collider graph in Figure R.2.
> 3. [Rationale behind choosing Random-fixed] - We have chosen to include Random-Fixed to maintain compatibility with CBED (Tigas et al. 2022).
>
> 4. [Number of interventions] - In Algorithm 2 we acquire data only from one intervention. The quantity 32 corresponds to the number of data samples drawn from the selected interventional distribution. We have also experimented with increasing this number (see Section 5.3). Those experiments show that increasing the number of samples to 1024 can significantly improve the results.
> 5. [Theorems in Appendix B] - the results are applicable to the GIT method. More specifically, Theorems 5-9 guarantee the “local convergence” of ENCO meaning that when acquiring enough real interventional data from a specific node parameters of all neighboring edges will converge and stop generating gradient. Based on this observation we conclude, in theorem 10, that if we choose any node that has a positive gradient magnitude, we will be able to discover more graph structure. We understand that the current proof is hard to follow. We will clarify the whole section in the camera-ready version.
>
> We again thank the Reviewer for the useful feedback. We hope that our answers clarify the issues. We would be happy to answer more questions, if needed, otherwise if everything is clear, we gently ask that the Reviewer consider raising the score.
>
> References:
> * Lowell, David, Zachary C. Lipton, and Byron C. Wallace. "Practical obstacles to deploying active learning." arXiv preprint arXiv:1807.04801 (2018).
> * Tigas, Panagiotis, et al. "Interventions, where and how? experimental design for causal models at scale." Advances in Neural Information Processing Systems 35 (2022): 24130-24143.
> * Ash, Jordan T., et al. "Deep batch active learning by diverse, uncertain gradient lower bounds." arXiv preprint arXiv:1906.03671 (2019).

---

> > ### Comment · Reviewer_E5qP · 2023-08-12
> >
> > I appreciate the authors' detailed response. I have revised the assigned score accordingly.

---

> > > ### Author Response · Authors · 2023-08-21
> > > **Thank you**
> > >
> > > We again thank the reviewer for useful questions and suggestions.

---

### Official Review · Reviewer_8Vao · 2023-07-06

**Soundness:** 3 good
**Presentation:** 3 good
**Contribution:** 3 good
**Rating:** 5
**Confidence:** 3

**Summary:**

This paper addresses the problem of targeting interventions to learn the causal graph. The ideal goal is to have the minimum number of interventions that is necessary to have identifiability.  The authors propose a Gradient-based Intervention Targeting method (GIT) that utilize the gradient estimator of a gradient-based causal discovery framework to provide signals for the intervention acquisition function. The paper includes extensive experiments on simulated and real-world datasets, demonstrating that GIT performs on par with competitive baselines, surpassing them in the low-data regime.


**Strengths:**

1.The paper addresses a significant problem in causality and machine learning, presenting a new approach to intervention targeting.
2. The proposed GIT method is model-agnostic and leverages the gradient estimator of a gradient-based causal discovery framework.
3. The authors provide extensive experiments on both simulated and real-world datasets, demonstrating the effectiveness of their approach, especially in low-data scenarios.
4. The paper is well written.


**Weaknesses:**

1. The paper could benefit from a more detailed explanation of the GIT method and its underlying principles. A summary of gradient based methods and why they work is also important to make the paper self-contained and easier to follow to people with a general background in causality and machine learning.
2. The authors could provide more insights into the limitations of their method, including potential biases and assumptions. Especially, as there is no solid theoretical justification for the convergence of the method. To my understanding, the convergence claim only holds with ENCO and with hard-to-interpret assumptions.


**Questions:**

1. The majority of the proofs in the appendix are just proof sketches, I think it is better to detail them so a more solid theory for the proposed method can be developed.
2. While the method is intuitively appealing and the extensive experiments provide very good results, is it possible to fail for some edge-case joint probability distributions?
3. Is there an empirical estimation of hte number of samples needed? Or a theoretical upper bound?
4. As neural causal discovery is more on the heuristic side (at least to my understanding), if the method badly fits the data distribution, how does the error propagate to GIT?
5. Do the author think (as a conjecture)  that GIT captures the minimum number of necessary interventions to have identifiability? Or do they think that there is a limitation to it, if so is it possible to give some examples?
6. Can the authors provide details for the assumption of theorem 5 to 10 (also please note that theorem 9 is missing in the appendix).


**Limitations:**

The empirical results are extensive and show great performance of the proposed algorithm. A more theoretical understanding of the method can provide more understanding to its limitations.

---

> ### Author Rebuttal · Authors · 2023-08-09
>
> We thank the Reviewer for the time and effort put into providing valuable feedback on our work. We are pleased to hear positive comments on the novelty of our work, the clarity of the manuscript, and the extensiveness of our experiments. We appreciate the constructive critique and questions which can help improve the publication. Below we provide answers to the Reviewer’s questions and concerns:
>
> 1. [More detailed explanation] - We agree with the Reviewer that a detailed description of the method benefits any paper. We feel that we put effort into explaining GIT in detail and would like to bring the Reviewer’s attention to the appropriate parts in the main body of the text: (a) intuition (lines 147-150) with grounding in existing literature (lines 91-98); (b) a pictorial representation of the GIT (Figure 1); (c ) formalism (lines 169-183); (d) assumptions and pseudo-code (lines 184-195); and (e) intuitive argument for theoretical justification (lines 196-205). Furthermore, in Section 3 we offer brief descriptions of two related gradient-based causal discovery methods: DIBS and ENCO (with more details postponed to Appendix C). We are committed to making the description as clear and easy to digest as possible, so if the Reviewer still feels that some details are missing, we kindly ask for suggestions, so we can address them accordingly.
>
> 2. [Insights into the limitations] - Thank you for this suggestion. We add the following limitations and future work section:
> * The theoretical grounding of the method involves multiple hard-to-interpret assumptions. Further work that simplifies the assumptions and identifies fail cases would benefit the community.
> * We provide proof that epsilon-greedy GIT converges with any causal discovery framework. As for pure GIT, we show its convergence only with the ENCO framework. The development of a more general theory that solidifies the approach is a promising future work direction.
> * Our method can be applied in the soft-intervention case, and providing appropriate experimental evaluation would be an interesting follow-up to this work.
> * Our method may need more interventions than the minimal number required to identify the causal structure. For example, GIT can be biased towards high-degree nodes, as interventions on them tend to affect a larger amount of structural parameters and result in larger gradients, which might cause suboptimal choices.
> We would also like to bring the Reviewer’s attention to the following insights described in the paper: the method's reliance on interventional batch size (Section 5.2) and on the number of graph samples (in Appendix F.4), a thorough analysis of the method's behavior (Section 5.3), and detailed discussions in Appendices F.5 and F.6.
>
> Questions:
> 1. [Proof sketches] - Thank you for raising this issue. We will clarify Appendix B in the camera-ready version and formalize GIT convergence theorem.
>
> 2. [Edge cases] - The proof in Appendix A shows that epsilon-greedy GIT has the same convergence guarantees as the underlying framework. Thus, for discussion on fail cases, we refer the Reviewer to the underlying causal discovery algorithm (for example, appendix B.2.4 in ENCO).
>
> 3. [Estimation of samples needed] - Yes, we experimented with different sizes of interventional batch size. For the specific graphs we tested, 1024 samples allow the method to intervene on each node at most once before SHD converges to 0 (see Section 5.3 and in Figure 9 in Appendix F). We found this result satisfactory to conclude that 1024 samples for the purpose of our experiments is enough.
>
> 4. [Error propagation] - Unsurprisingly, GIT is susceptible to the errors of the underlying methods. Due to a lack of proper theoretical frameworks, these are hard to quantify. However, our positive empirical results, let us have an optimistic belief that there is a significant resilience to these errors. We conjecture that this might be due to the fact that the probabilistic nature of the causal frameworks may, on average, shield us from the worst cases scenarios.
> We also identified a case, in which the errors might be hard to overcome. Consider the collider graph from Figure 7 in the Appendix. The collider graph is difficult to learn because it is hard to model precisely conditional distribution in the collider node. Our method, together with BOED, struggles to extract useful information from the model in this example as they are not able to match the Random baseline.
>
> 5. [Required number of interventions] - Thank you for this question. We do not expect that GIT needs only a minimum number of interventions. We add this to the limitation list. This is for two reasons: the underlying framework may not have this property (which is, for example, the case of ENCO), and GIT might not optimally choose the required interventions. On the positive side, our empirical results clearly demonstrate that GIT performs well and in practice outperforms existing methods.
>
> 6. [Assumptions details] - The theorems 5-10 are a direct adaptation of the convergence results for the ENCO method. In consequence, apart from the observations 1-3 mentioned in our Appendix B, we also have to assume all the requirements made by ENCO - we refer to sections B.1 of the ENCO Appendix (Assumptions 1-5) and the explained intuition behind them in ENCO Appendix B.2.1. and B.2.2. We understand that the current proof is hard to follow. We will clarify the whole section in the camera-ready version.
>
> We again thank the Reviewer for the valuable feedback. Should you have any other questions or concerns, please let us know. If the answers are satisfactory, we gently ask the Reviewer to consider raising the score.
>
> References:
>
> [ENCO] Phillip Lippe et al. Efficient neural causal discovery without acyclicity constraints. 2021.
>
> [DIBS] Lars Lorch et al. Dibs: Differentiable bayesian structure learning. NeurIPS, 2021.

---

> > ### Comment · Reviewer_8Vao · 2023-08-12
> >
> > Thank you for your response!
> >
> > However, I believe the current score honestly reflect the quality of this work and will keep my score for now.

---

> > > ### Author Response · Authors · 2023-08-17
> > > **Thank you**
> > >
> > > Thank you again for many useful comments. If you have any further suggestions or questions, please let us know.

---

### Official Review · Reviewer_qbh5 · 2023-07-26

**Soundness:** 3 good
**Presentation:** 4 excellent
**Contribution:** 3 good
**Rating:** 8
**Confidence:** 1

**Summary:**

The paper studies the problem of efficiently inferring causal structure from data. Observational data often falls short in providing a full picture of the causal structure, while obtaining interventional data tends to be costly. Therefore, optimizing the collection of interventional data to reduce the number of necessary experiments becomes crucial. In this context, the authors present Gradient-based Intervention Targeting (GIT), a novel approach to enhancing causal discovery from data. Uniquely, GIT takes advantage of gradient estimators from gradient-based causal discovery frameworks, paving the way for efficient intervention targeting. Thanks to its plug-and-play nature, GIT is compatible with various frameworks. The authors validate the effectiveness of GIT via extensive tests on synthetic and real-world datasets, demonstrating its superior performance, especially in low-data scenarios.

**Strengths:**

-	The problem studied in this paper is important.
-	The paper is written in a very clear way. The background, method, and empirical results are all presented nicely.
-	The proposed method demonstrates very good empirical performance.


**Weaknesses:**

Typos:

-	In the final sentence of the first page, the in-text citation should use \citep instead of \citet.
-	On line 156, there appears to be a large space between 'A' and the period.


**Questions:**

NA

---

> ### Author Rebuttal · Authors · 2023-08-09
>
> We thank the Reviewer for positive feedback on our work. We are glad that the Reviewer considers the studied problem to be important and that they appreciate a very good empirical performance of our method. Is it good to hear that we managed to clearly describe our work and present the results nicely.
>
> We thank the Reviewer for identifying the typos and will correct the typos in the camera-ready version of the paper.

---

### Official Review · Reviewer_6LvC · 2023-07-27

**Soundness:** 4 excellent
**Presentation:** 4 excellent
**Contribution:** 3 good
**Rating:** 7
**Confidence:** 3

**Summary:**

The authors present an approach for inferring the most informative intervention targets for the task of causal discovery; where they propose an alternative to bayesian experimental design approaches. They use the gradients of the casual discovery algorithm's loss function to find the intervention targets, which helps the algorithm to learn about regions of the graphs that it is most uncertain about. They experiment with the ENCO causal discovery algorithm and show how their approach for interventional target acquisition outperforms other approaches on both synthetic and real data benchmarks.

**Strengths:**

* Since we can only learn the causal graphs up to an equivalence class using observational data, it is important to collect interventional data in an efficient manner for causal discovery. Hence, the authors consider a very relevant problem in the area.

* The proposed approach is novel to the best of my knowledge and also technically sound with extensive experimentation to back it. The authors compare a good set of baselines and perform several ablation studies to understand the effectiveness of their approach.

* The paper is very well written with good presentations of the experiment results and details of the proposed approach.

* The proposed approach could be especially significant for large graphs with more nodes, where estimators for mutual information might not work well and the approaches based on them could suffer.

**Weaknesses:**

I do not think the paper has any serious weaknesses, but I have listed some of them in the questions section ahead. One suggestion for the authors would be to test their approach with synthetic data containing more nodes, as it could help us understand how the proposed intervention target acquisition approach scale with the size of the graph.

**Questions:**

* The approach works with causal discovery algorithms that maintain a distribution over the structure of the graph. Is that assumption limiting in some manner? What about approaches where we do not maintain a distribution over graphs, could the approach be applied with a single-point estimate for the graph?

* A related question to the previous one, how does the approach perform when we change the number of graphs sampled from the distribution over graphs? Are there ablations studies that test the effect of changing $|G|$ on the performance of the causal discovery algorithm?

* The authors consider only single-node hard interventions for target acquisition; what are the reasons for not considering soft interventions? Is it a limitation of the proposed approach or soft interventions are not as informative as hard interventions for the task of causal discovery?

**Limitations:**

Yes, the authors have addressed any potential negative societal impact of their work.

---

> ### Author Rebuttal · Authors · 2023-08-09
>
> We thank the Reviewer for a very encouraging review. We are glad that the Reviewer appreciated the importance of the problem, novelty, soundness of the approach and experiments, and clear presentation.
>
> Below we answer the specific questions asked by the Reviewer:
>
> - [Scalability of the method]: We thank the Reviewer for suggesting studying scalability. For the rebuttal, we ran an additional experiment on the jungle graph with 100 nodes, in which we compared different acquisition methods used with ENCO (see Figure R.1 in the additional rebuttal PDF). We can indeed observe that GIT significantly outperforms Random and MI-based approaches given the intervention budgets as in our main text experiments. For the camera-ready version, we will prepare a section about scalability, with results on other synthetic graphs.
>
> - [Maintaining distribution over graphs]: In order to be compatible with GIT, the underlying causal discovery method needs to maintain a distribution over the graphs (causal DAGs). This assumption is explicitly mentioned in the paper, in Section 4, paragraph “Requirements for causal discovery algorithm A”. That being said, we note that the assumption is met for a broad class of recent causal discovery algorithms, including ENCO [1], DIBS [3], SDI [2], DCDI [4], and DECI [5]. In fact, we are not aware of any modern neural network-based approach for which this requirement would not be satisfied.
>
> - [Impact of number of graphs sampled]: We performed the mentioned ablation and reported the results in Figure 10 in Appendix. We varied the number of sampled graphs from 10 to 70 and observed no significant impact on the results. In the experiments in the main text, we always use 50 samples.
>
> - [Using soft interventions]: We thank the Reviewer for this interesting question. The main reason we considered hard interventions is easier comparability with prior works such as [1], which used hard interventions in their experiments. However, this is not a limitation of our method, and there is no reason for soft interventions not to work with GIT. We leave an empirical evaluation in such a scenario for future work.
>
>
> We would like to express our gratitude once more for the Reviewer's positive rating and constructive feedback, which have contributed to the improvement of our paper.
>
> [1] P. Lippe, T. Cohen,  E. Gavves. Efficient neural causal discovery without acyclicity
> constraints. arXiv:2107.10483.
>
> [2] N. R. Ke, O. Bilaniuk, A. Goyal, S. Bauer, H. Larochelle, B. Schölkopf, M.  Mozer, C. Pal, and Y. Bengio. Learning neural causal models from unknown interventions, arXiv:1910.01075.
>
> [3] L. Lorch, J. Rothfuss, B. Schölkopf, A. Krause. Dibs: Differentiable bayesian structure learning. Advances in Neural Information Processing Systems 34, 2021.
>
> [4] Ph. Brouillard, S. Lachapelle, A. Lacoste, S. Lacoste-Julien, A. Drouin. Differentiable causal discovery from interventional data. Advances in Neural Information Processing Systems, 33, 2020.
>
> [5] T. Geffner, J. Antoran, A. Foster, W. Gong, Ch. Ma, E. Kiciman, A. Sharma, A. Lamb, M. Kukla, N. Pawlowski, M. Allamanis, Ch. Zhang. Deep End-to-end Causal Inference. arXiv:2202.02195

---

> > ### Comment · Reviewer_6LvC · 2023-08-16
> >
> > Thanks for the good response during the rebuttal! My concerns are addressed and I think my original rating is still fair enough with regards to the submission.

---

> > > ### Author Response · Authors · 2023-08-17
> > > **Thank you**
> > >
> > > Again, thanks for your work towards making our paper better.

---

### Author Rebuttal · Authors · 2023-08-09

We would like to thank all the Reviewers for taking the time to review our work and providing us with insightful feedback. We are glad that the Reviewers appreciate the strengths of our paper, including clear writing and good presentation (Reviewers 6LvC, qbh5, 8Vao, E5qP), extensive experimental results (Reviewers 6LvC, qbh5, 8Vao, E5qP), significance of the studied problem (Reviewers 6LvC, qbh5, 8Vao, E5qP), and novelty of our approach (Reviewers 6LvC, 8Vao). It is good to hear that our study has no serious weaknesses (Reviewer 6LvC), is technically sound (Reviewers 6LvC, zVWz), and presents a clean solution (Reviewer E5qP).

We provide specific answers to the Reviewers' individual concerns posted as separate comments below and attach a pdf file with figures referenced in individual responses.

---

> ### Comment · Area_Chair_xL9S · 2023-08-17
>
> Thank you for your note. We'll take it into consideration.

---

### Decision · Program_Chairs · 2023-09-21

**Decision:**

Accept (poster)

**Comment:**

The paper introduces a novel approach called Gradient-based Intervention Targeting (GIT) for efficiently selecting intervention targets in the context of causal discovery. It utilizes gradient information to identify the most informative intervention targets, with a focus on minimizing the number of interventions required for causal graph learning. The authors extensively validate GIT through experiments on both synthetic and real-world datasets, showcasing strong performance, particularly in low-data scenarios. There were some questions and concerns raised by reviewers including the performance of the baselines and how partial information is added. The paper would benefit from addressing these comments.